

# High sensitivity of simulated fog properties to parameterized aerosol activation in case studies from ParisFog

Pratapaditya Ghosh[1,2], Ian Boutle[3], Paul Field[3,4], Adrian Hill[3,5], Anthony Jones[3], Marie Mazoyer[6], Katherine J Evans[7], Salil Mahajan[7], Hyun-Gyu Kang[7], Min Xu[7], Wei Zhang[7], Noah Asch[1,2], and Hamish Gordon[8,2]

[1]Department of Civil and Environmental Engineering, Carnegie Mellon University, 5000 Forbes Avenue, Pittsburgh, 15213, United States
[2]Center for Atmospheric Particle Studies, Carnegie Mellon University, 5000 Forbes Avenue, Pittsburgh, 15213, United States
[3]Met Office, Fitzroy Road, Exeter, EX1 3PB, United Kingdom
[4]School of Earth and Environment, University of Leeds, Leeds, LS2 9JT, United Kingdom
[5]European Center for Medium-Range Weather Forecasting, Reading, UK
[6]CNRM, Université de Toulouse, Météo-France, CNRS, Toulouse, France
[7]Oak Ridge National Laboratory, Oak Ridge, TN, 37831, USA
[8]Department of Chemical Engineering, Carnegie Mellon University, 5000 Forbes Avenue, Pittsburgh, 15213, United States

**Correspondence:** Hamish Gordon (gordon@cmu.edu)

**Abstract.** Aerosols influence fog properties such as visibility and lifetime by affecting fog droplet number concentrations ($N_d$). Numerical weather prediction (NWP) models often represent aerosol-fog interactions using highly simplified approaches. Incorporating prognostic size-resolved aerosol microphysics from climate models could allow them to simulate $N_d$ and aerosol-fog interactions without incurring excessive computational expense. However, microphysics code designed for coarse spatial

resolution may struggle with sub-kilometer-scale grid spacings. Here we test the ability of the UK Met Office Unified Model to simulate aerosol and fog properties during case studies from the ParisFog field campaign in 2011. We examine the sensitivity of fog properties to variations in $N_d$ caused by modifications to simulated aerosol activation.

Our model with 500 m horizontal resolution and interactive aerosol and cloud microphysics significantly underpredicts $N_d$, although only slightly underestimates the cloud condensation nuclei concentration. With an updated version of the Abdul-

Razzak and Ghan (2000) activation scheme, we produce $N_d$ that are more consistent with those predicted by a cloud parcel model under fog-like conditions. We activate droplets only by adiabatic cooling. We incorporate more realistic hygroscopicities for sulfate and organic aerosols and explore the sensitivity of simulated $N_d$ to unresolved updrafts. We find that both $N_d$ and simulated fog liquid water content are very sensitive to the updated activation scheme but remain unaffected by the update to hygroscopicities. Our improvements offer insights into the physical processes regulating $N_d$ in stable conditions, po-

tentially laying foundations for improved operational fog forecasts that incorporate interactive aerosol simulations or aerosol climatologies.





# 1 Introduction

Fog events are associated with reduced visibility that affects road, water and air transportation. Behind freezing conditions, fog is the second leading cause of weather delays and cancellations at major airports across the United States (Goodman and Griswold, 2019). At the Paris-Charles-de-Gaulle airport, the frequency of takeoffs and landings is reduced by half when visibility drops below 600 m due to fog (Roquelaure and Bergot, 2009). Therefore, accurate operational weather forecasts for fog are necessary.

Aerosol particles activate to form fog droplets, affecting visibility and fog life cycle (Stolaki et al., 2015; Boutle et al., 2018; Poku et al., 2019; Wainwright et al., 2021; Yan et al., 2021; Poku et al., 2021; Duplessis et al., 2021). These droplets in turn can influence light extinction, sedimentation and evaporation rates (Ackerman et al., 2004; Hill et al., 2009) based on their concentration. Therefore, numerical weather prediction models are starting to use interactive prognostic aerosol microphysics schemes to improve the representation of aerosols and therefore better forecast fog (Jayakumar et al., 2021). In climate models, it is important to simulate droplet concentrations accurately in low-level clouds, which include different types of fog (for example radiation fog or advection fog), in order to simulate the indirect effects of aerosols on climate. Since weather and climate models are becoming increasingly unified, consistent representations of aerosol-fog interactions that will work in both weather and climate models are desirable. If we can demonstrate that a comprehensive prognostic aerosol scheme can improve simulations, it would then be possible to test possible simplifications that would save computational expense for specific operational settings, for example using aerosol climatologies.

However, while low clouds have much in common with fog, there are also important differences, especially in how aerosols affect clouds and fog. For the same aerosol concentration, fog droplet concentrations ($N_d$) are expected to be lower than droplet concentrations in clouds. The main physical reason for the low $N_d$ is that in fog, updraft speeds are generally lower than in clouds, reducing aerosol activation rates. Activation is likely to occur at least in part due to radiative cooling (Gultepe et al., 2007; Wærsted et al., 2017; Poku et al., 2021) rather than vertical air motion.

In this study, we improve the simulated fog droplet number concentration in the UK Met Office Unified Model (UM) for fog cases observed near Paris in 2011. We are inspired by several previous specific studies: for example, using a single column model, Stolaki et al. (2015) found that doubling the number of cloud condensation nuclei concentration (CCN) in a subset of the days we use in our case study from the ParisFog campaign led to a 60% increase in the liquid water path. Simulating actual supersaturation and condensational growth of aerosols in a large eddy simulation (LES) model, Boutle et al. (2018) found that the onset of well-mixed fog (in which the vertical profile of liquid water content (LWC) is near adiabatic) was dependent on aerosol activation. The interaction of the land surface with radiation led to the initial fog formation, while the fog layer itself started interacting with radiation only when a substantial fraction of the population of the largest aerosols activated, resulting in optically thicker fog. WRF-Chem modeling studies by Yan et al. (2021) and Jia et al. (2019) found that an increase in the number of cloud condensation nuclei concentrations could advance fog formation, delay fog dissipation (by about an hour), enhance fog intensity and generate long and severe fog events. An LES study by Vié et al. (2024) found significant improvements in simulations of fog in the UK could be achieved by modifying the cooling rates used in aerosol activation,





which were previously overestimated. The high sensitivity of fog properties, including even theermodynamic profiles, to the simulated cooling rates used in the activation scheme also highlights the importance of aerosol-fog interactions.

Weather and climate models that represent aerosols prognostically (e.g. Jayakumar et al., 2021; Lohmann, 2002) use aerosol activation parameterizations as the source of cloud and fog droplets. The parameterizations typically balance a source of super-saturation, usually only adiabatic cooling due to updrafts, with a sink due to condensation on activating aerosol particles. The parameterizations are needed because changes in supersaturation generated by updrafts typically occur on timescales of a few seconds, too short to be explicitly resolved by large-scale models. The widely used Abdul-Razzak and Ghan (2000) activation scheme, henceforth 'ARG', was originally developed for boundary layer stratiform clouds and can also be applied for cumulus clouds (Ghan et al., 2011). It calculates $N_d$ from cooling due to vertical air motion, temperature, the aerosol size distribution, and their hygroscopicity. Similarly to most climate models, many cloud-resolving scale modeling studies, including those of fog, use the ARG activation scheme (e.g. Chapman et al., 2009; Jia et al., 2019). However, the parameterization of visibility in numerical weather prediction models is sometimes an exception in that the ARG scheme is not always used. In the UM, for example, visibility is parameterized as a function of aerosol mass loading and relative humidity, via a scheme that is similar to a simplified diagnostic representation of aerosol activation (Clark et al., 2008) that does not depend on vertical air motion. However, this parameterization assumes that the aerosols are monodisperse. If possible, it would be preferable to calculate the visibility of the prognostic $N_d$ and LWC from a cloud microphysics parameterization that handled all types of cloud, including fog, with a consistent activation scheme that takes into account aerosol properties such as size and hygroscopicity.

To represent the cloud microphysics of fog in a weather prediction or climate model, several other assumptions are also needed. One important assumption is that the fog size distribution has a single droplet mode, as these models generally have 'bulk microphysics schemes' without size sections or multiple modes for cloud droplets. However, two droplet modes, both below 30 $\mu$m diameter, are frequently observed during fog (Mazoyer et al., 2022). The error in approximating the fog droplets with a single mode remains to be quantified. Related to this is the assumption that unactivated haze aerosols and droplets are distinct entities. The wet critical diameter (threshold diameter of a wet particle beyond which it can grow spontaneously and form a droplet) can be around 3-5 µm (Mazoyer et al., 2022), a factor of 10 greater than the dry critical diameter (threshold diameter of a dry aerosol before condensation, beyond which spontaneous growth happens) (Mazoyer et al., 2019). Bulk aerosol microphysics coupled to bulk cloud microphysics schemes are well known to struggle to fully capture the details of the activation and deactivation of haze (Yang et al., 2023), long recognized as a difficult problem in cloud modeling (e.g. Árnason and Brown, 1971). This is especially challenging without representing complex mixing interactions or accounting for surface tension changes due to surfactants. A key aim of this paper is to examine whether bulk microphysics is nonetheless a sensible pragmatic solution to modeling aerosol-fog interactions despite these shortcomings.

In addition to these assumptions, shorter timesteps are typically required to maintain numerical stability as grid resolutions become finer. In large eddy simulations of fog with timesteps smaller than 10s, Thouron et al. (2012) and Schwenkel and Maronga (2019) found that the frequently applied saturation adjustment approximation, which assumes supersaturations are removed in one timestep, resulted in overestimated $N_d$. In LES simulations with an 0.1s timestep, Schwenkel and Maronga (2019) find $N_d$ is 60% higher than when it was simulated without assuming saturation adjustment. However, in lower-resolution



models with timesteps of around 30 s or longer, saturation adjustment is unlikely to introduce large biases unless the fog is very thin, such that the sink of water vapor to droplets and aerosols is very small and the timescale for saturation to reduce to zero exceeds the model timestep (Thouron et al., 2012; Gordon et al., 2020). For example, in a fog with $N_d = 100\,\mathrm{cm}^{-3}$ and a droplet radius of 5 μm, the relaxation timescale would be around 14 s (Politovich and Cooper, 1988; Grabowski and Wang, 2013).

Here we use a regional configuration of the Unified Model to investigate aerosol activation in case studies of fog over Paris. Similar simulation systems are used for the operational forecasting of fog over London (Boutle et al., 2016) and Delhi (Jayakumar et al., 2021), although the published systems did not use double-moment cloud microphysics. We use observational data from the ParisFog field campaign (Haeffelin et al., 2010) at the Instrumented Site for Atmospheric Remote Sensing Research (SIRTA) in November 2011. The occurrence of 11 fog cases in two weeks with detailed observations of aerosol and fog microphysics makes the campaign preferred for our study (although other field campaigns such as LANFEX in the UK (Price et al., 2018) or WiFEX in New Delhi (Ghude et al., 2017) could also have been used). In this paper, Part I, we simulate fog $N_d$ using adiabatic cooling as a source of supersaturation, in a 500 m resolution simulation with double-moment aerosol and cloud microphysics, nested inside lower-resolution simulations. We focus on improving the parameterization of activation without changing the fundamental mechanism of supersaturation generation by adiabatic cooling. Although it is likely that adiabatic cooling is not the dominant source of supersaturation in all the fog cases we study, adiabatic cooling is the simplest and most widely used approach to activation in models.

We evaluate simulated aerosol and fog droplet concentrations using the ParisFog observations and test the existing activation scheme in our model. We discuss the observational data and details of several instruments in Section 2. The setup of our model is discussed in Section 3. Section 4 explains our model developments that affect $N_d$ in fog. In Section 5, we discuss the model evaluation and the results. Finally, in Section 6, we present our conclusions.

In the companion paper we add radiative cooling as a source of supersaturation in the ARG scheme and study the fog lifecycle simulated in the model. We also design several sensitivity studies to address missing sinks and model artefacts that might affect the droplet budget in the model. Finally, we calculate the relative importance of the adiabatic and radiative cooling components of activation in the droplet budget.

## 2   Measurements and Case Study

In this study, we use measurements from the ParisFog field campaign to evaluate our model. ParisFog refers to several field campaigns that have taken place at the Instrumented Site for Atmospheric Remote Sensing Research (SIRTA) since 2006 to understand the lifecycle of fog. Haeffelin et al. (2005) describe the SIRTA observatory (48.713° N, 2.208° E), located near the Ecole Polytechnique Palaiseau, on the Saclay plateau at 160 m above sea level about 20 km southwest of Paris city center, which is mostly around 35 m above sea level. The site is located in a semi-urban environment composed of roughly equal portions of agricultural fields, wooded areas, housing, and industrial developments. Prevailing winds advect clean air from the





Atlantic Ocean towards the site. Winds from the North-East carry pollutants from the Paris metropolitan area, affecting the aerosol composition over SIRTA.

We show observations of 11 fog events between 15 and 25 November 2011 in this paper. Mazoyer et al. (2019) categorizes the four fog events on 16 (afternoon), 24 (morning and afternoon) and 25 November as stratus lowering, and the remaining events on 15, 16 (morning), 18, 19, 21, 22, 23 November as radiation fogs. The microphysical properties of the droplets and aerosols were measured regularly using different instruments. The details of all the instruments and their setup used during the ParisFog campaign are well documented in previous studies (Hammer et al., 2014; Stolaki et al., 2015; Elias et al., 2015; Dupont et al.,

2016; Mazoyer et al., 2017, 2019, 2022); those we use here are listed in Table 1. To evaluate simulated meteorology, we use near surface temperature and relative humidity data from the weather sensors at SIRTA (48.7° N, 2.2° E) and Trappes (48.7° N, 2° E). We also use a regional weighted average of measurements from Trappes, Orly (48.7° N, 2.4° E), and Paris Montsouris (48.8° N, 2.3° E), following 3.2.1 of (Chiriaco et al., 2018). We also use the vertical profiles of temperature and relative humidity from radiosondes launched by Météo-France twice a day from Trappes (48.7° N, 2° E), situated 15 km northwest of

the SIRTA observatory.

**Table 1.** Instruments used to measure optical, radiative and microphysical properties of aerosols and droplets during the ParisFog field campaign that we also use in our model evaluation.

| Instrument Name | Measured Parameter | Diameter | Resolution |
|---|---|---|---|
| Scanning Mobility Particle Sizer (SMPS) | Dry Aerosol Size Distribution | 10.6–496 nm | 5 min |
| Palas Welas-2020 Particle Counter (WELAS) | Ambient aerosol and droplet size distribution | 0.39–42 μm | 5 min |
| DMT Fog Monitor (FM-100) | Droplet number size distribution | 2–50 μm | 1 min |
| Radiosondes | Vertical temperature and RH profile | NA | NA |
| Weather Station and Sensors | Near-Surface Temperature and RH | NA | 1 min |

The dry aerosol number size spectrum was measured using a Scanning Mobility Particle Sizer (SMPS). The instrument was placed inside a shelter. Air was sampled and passed through an aerodynamic size discriminator PM2.5 inlet and then through a dryer, reducing the relative humidity to less than 50%. The differential mobility analyzer (DMA; TSI 3071) in the SMPS that selected particles from 10.6 to 496 nm in diameter had hydrophilic filters lowering the relative humidity to less than 30%

(Denjean et al., 2014), the point at which point we can neglect water uptake by typical urban aerosols (e.g. Seinfeld and Pandis, 1998). The size distribution of larger particles at ambient relative humidity was measured using two optical spectrometers: the WELAS-2000 (Palas GmbH, Karlsruhe, Germany) and the fog monitor FM-100 (Droplet Measurement Technologies Inc., Boulder, CO, USA). The WELAS-2000 (hereafter referred to as WELAS) is designed to measure particles within 0.4 and 40 μm in diameter, but previous studies found that the detection efficiency decreases drastically below 1 μm (Heim et al., 2008;

Elias et al., 2015). Hammer et al. (2014) used data with diameters larger than 1.4 μm and Mazoyer et al. (2019) used 0.96 μm as a threshold diameter. The FM-100 fog monitor provides droplet size distribution from 2 to 50 μm in diameter, thus having a large overlap with the WELAS. However, comparisons by Burnet et al. (2012) and  Elias et al. (2015) show that distributions



from these instruments overlap only from 5-9 μm diameter, because the FM-100 overestimated particles lower than about 5 μm while the WELAS underestimated droplets larger than 10 μm (Burnet et al., 2012; Elias et al., 2015). Figures 1 and A1 of

Mazoyer et al. (2019) demonstrate the bias. In our work, we plot the overlapping range from both instruments. The FM also provides LWC in the fog.

## 3 Model Description

### 3.1 Meteorological Model

We use UM version 13.0 to study aerosol activation and aerosol-fog interaction. To enable prognostic aerosol and cloud mi-

crophysics, simulations are run within the NUMAC (Nested Unified Model with Aerosols and Chemistry) system (Gordon et al., 2023). The coupling of the double moment cloud microphysics (Field et al., 2023) to the double moment aerosol microphysics (Mulcahy et al., 2020) is described by Gordon et al. (2020). We use three different configurations of the model: a global model, a 4 km grid resolution regional model nested within it, and the 500 m grid resolution regional model, nested within the 4 km model. The regional models use time steps of 120 seconds and 30 seconds for the dynamics, respectively, with

a common chemistry time step of 120 seconds.

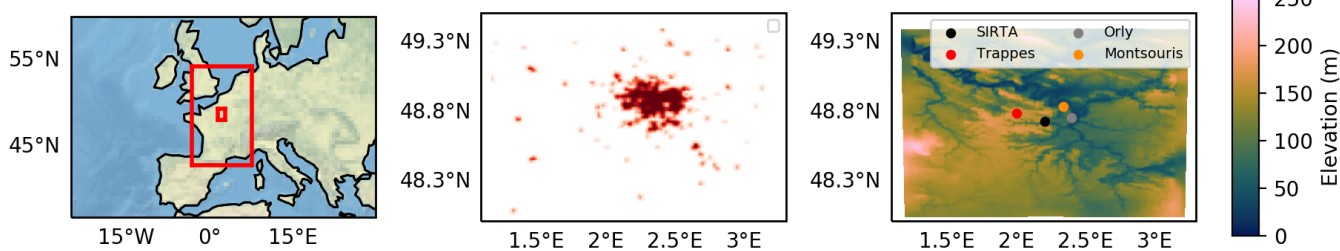

**Figure 1.** Subfigure (a): Location of the two regional nested domains used in this study is shown. Both the domains are centered over the SIRTA observatory near Paris. The outer domain has 4 km grid resolution and the inner domain has 500 m resolution. Both the domains have 300 grid points in the latitude and longitude directions. Subfigure (b): Urban grid cells in the 500 m resolution model domain. The regional scale is set by Paris near the center and Chartres in the south-west at approximately 48.44° N, 1.49°E. Subfigure (c): Surface altitude (m) in the 500 m resolution domain is shown here. Measurement sites at SIRTA (observatory, domain center), Trappes (radiosondes), Orly and Paris Montsouris (weather stations) are shown in dots.

This work focuses primarily on results from the regional 500 m resolution model. The 500 m model is centered over the SIRTA observatory as shown in Figure 1(a). Subfigure (b) shows the urban grid cells in the 500 m resolution model (roughly the area of Paris city) and subfigure (c) shows surface elevation (m) in our 500 m resolution domain. The SIRTA observatory (the domain centre), Trappes (location of radiosondes, discussed later), Orly, and Paris Montsouris (weather stations, discussed

later) are shown in dots. We use 300 grid points in both latitude and longitude directions in both our 500 m and 4 km-resolution



domains, with 70 vertical levels extending up to 40 km (61 levels are below 18 km, and 16 levels below 1000 m). These domains use the Regional Atmosphere and Land Version 3 (RAL3) physical atmosphere model configuration (Bush et al., 2024), which is similar to Regional Atmosphere and Land Version 2 (Bush et al., 2022) but with the two moment Cloud–AeroSol Interacting Microphysics (CASIM) scheme (Shipway and Hill, 2012; Grosvenor et al., 2017; Field et al., 2023) used by default

in place of the single moment microphysics scheme of Wilson and Ballard (1999). In section 3.2 we describe the microphysics schemes in detail. The RAL3 configuration also uses the more recently developed cloud fraction scheme of Weverberg et al. (2021) by default, however we found that the cloud cover was underpredicted with this scheme, and this prevented a robust model evaluation of cloud properties. The underprediction of cloud cover in the bimodal scheme is discussed in more detail by Van Weverberg and Morcrette (2022), who compare all the cloud schemes available in the UM. Therefore in this work we

used the sub-grid cloud fraction scheme by Smith (1990), which is better able to generate cloud when supersaturations are low, as in fog. The grey-zone turbulence parameterization of Boutle et al. (2014) is used to represent subgrid-scale mixing in the boundary layer within the boundary layer scheme described by Lock et al. (2000). The model uses the ENDGAME semi-Lagrangian dynamical core (Wood et al., 2014; Thuburn, 2016) and no parameterized convection. Radiative transfer is treated by the SOCRATES scheme (Manners et al., 2015) based on work by Edwards and Slingo (1996).

The regional simulations use one-way nesting, which means the global model is independent of the regional model, and the 4 km model is independent of the 500 m-resolution model. Lateral boundary conditions for the 500 m model are provided by the 4 km-resolution model, then the global model provides the lateral boundary conditions for the 4 km model. The 4 km model is configured identically to the 500 m model. It covers all of France and some parts of the UK as shown in Figure 1. The global model has a horizontal resolution of $1.87° \times 1.25°$ (labeled N96 within the UM framework) with 85 vertical levels

extending up to 85 km from the surface. It uses the same code base as the regional model, but because the global model uses the global atmosphere science configuration GA7.1 (Walters et al., 2019), which differs from the RAL3 configuration, there are differences between the settings of various parameters in the simulations of dynamics, boundary layer, radiation, and sub-grid cloud. The most important difference for this work is that the configuration stipulates that the PC2 (Prognostic Cloud, Prognostic Condensate) sub-grid cloud scheme (Wilson et al., 2008) be used in the global model instead of the Smith

or bimodal schemes, and convection is parameterized in the global model.

To ensure the simulated meteorology closely follows reality, the regional model is initialized at 12:00 UTC on each day from 14[th] Nov to 25[th] Nov and run for 28 hours each time. We used the UK Met Office global weather forecast fields, available at high resolution (~ 12 km) to initialize meteorological variables such as temperature, wind speed, and humidity in the nested regional models. These global weather forecasts closely follow real meteorology because they are generated with data

assimilation, but we do not use data assimilation in our own simulations, to avoid complicating the interpretation of our results. The concentrations of chemistry and aerosol species are not represented in the global forecast, so are carried forward from the previous forecast as described by Gordon et al. (2023). The aerosol and chemistry fields at the start of the simulation on 14 November are initialized from a nudged global simulation with a configuration similar to the atmosphere-only UKESM1 model (Mulcahy et al., 2020; Sellar et al., 2020).





Results from the first 4 hours of each forecast simulation are discarded to allow simulated low clouds and fog in the initial-
ization files to adjust to the higher spatial grid resolution and aerosol concentrations in our model. We expect that 4 hours is an
adequate spin-up time, first because fog is rarely observed during the spin-up period (12:00-16:00 UTC), and second because,
even when it is present, we expect that $N_d$ at any given time are independent of $N_d$ four hours earlier. We acknowledge that
this may not be enough time to fully represent adjustments of cloud liquid water path to aerosol (e.g. Glassmeier et al., 2021).

## 3.2    Chemistry, aerosol, and cloud microphysics

We use the StratTrop chemistry scheme within the UK Chemistry and Aerosol (UKCA) submodel (Archibald et al., 2020;
Gordon et al., 2023) in this study. 84 chemical species are treated as prognostic tracers and there are 291 chemical reactions.
This chemistry mechanism is relatively complex and expensive compared to, for example, the simpler alternative available in
UKCA that uses climatological oxidant fields, but it is currently required to simulate nitrate aerosols, which are an important
component of total aerosol mass in Paris (Crippa et al., 2013; Roig Rodelas et al., 2019). Several global modeling studies have
used this scheme over the last five years (e.g. Sellar et al., 2019, 2020; Stevenson et al., 2020; Thornhill et al., 2021). In the
regional models, the most important difference arises from the lack of a convection parameterization, which means NO is not
produced from the lightning by default. To address this problem, the lightning flash rate parameterization of McCaul et al.
(2009) was used in our study, as in Gordon et al. (2023). Updates to the simulated sulfur cycle described by Mulcahy et al.
(2023) are included.

This study uses the double moment modal Global Model of Aerosol Processes (GLOMAP-mode, hereafter referred to as
GLOMAP) aerosol microphysics scheme (Mann et al., 2010; Mulcahy et al., 2020) within UKCA including the improvements
of Mann et al. (2012) and the changes recommended by Mulcahy et al. (2018). Different aerosol species are represented by
5 log-normal modes labeled nucleation soluble, Aitken soluble, Aitken insoluble, accumulation soluble, and coarse soluble
by Mann et al. (2010). In practice, the difference between 'soluble' and 'insoluble' modes is that soluble modes can grow by
taking up water, and be wet scavenged, while 'insoluble' modes cannot. Mineral dust is treated separately by the CLASSIC
(Coupled Large-scale Aerosol Simulator for Studies In Climate) sectional scheme of Woodward (2001). All aerosol and most
chemical species are transported by the Unified Model's semi-Lagrangian advection scheme. For the first time in the UM
regional modeling configurations, our work also includes ammonium and nitrate aerosols along with black carbon, organic
carbon, sea-salt and sulfate, following Jones et al. (2021). These chemical species are assumed to be internally mixed within
the lognormal modes. Fossil fuel, biofuel, and biomass burning emissions of black carbon and organic carbon are emitted to
the Aitken insoluble mode. Once 10 monolayers of sulfate or secondary organic aerosols have condensed on these primary
particles, they are moved to the 'soluble' modes. The mass and number concentration of these aerosol species in each mode
are independent prognostic variables, which is key to accurately representing aerosol-cloud interactions (Bellouin et al., 2013).
Each mode has a fixed 'width' or geometric standard deviation but the median diameter is calculated from mass and number
concentrations. The width of the accumulation mode is set to 1.4, based on a comparison with a sectional model and a review
of the observational literature by Mann et al. (2012).





The Coupled Model Intercomparison Project (CMIP6) inventory (Feng et al., 2020) is used for anthropogenic and natural aerosol emissions in the global model. The high resolution Emission Database for Global Atmospheric Research developed to assess Hemispheric Transport of Air Pollutants (EDGAR-HTAP) is used for anthropogenic emissions in the regional model (Janssens-Maenhout et al., 2015). EDGAR-HTAP provides monthly global anthropogenic emissions at $0.1° × 0.1°$ horizontal resolution (for 2010). Non-anthropogenic and biomass burning emissions for the regional model are not included in the EDGAR inventories, so are taken from the CMIP6 inventories used by the global model instead. These datasets are used after regridding to the $1° × 1°$ resolution of the global model, to avoid additional processing.

Different cloud microphysics schemes are used for different configurations in our study. The single-moment scheme of Wilson and Ballard (1999) is used in the global model. This scheme only uses the mass of hydrometeors as prognostic variables. However, we use the two moment Cloud–AeroSol Interacting Microphysics (CASIM) scheme (introduced earlier) in our regional domains to simulate the lifecycle of fog. In CASIM, cloud, rain, ice, snow, and graupel are represented by separate gamma distributions. The mass and number concentration of each species are prognostic variables. We follow Field et al. (2023) in using a shape parameter of 2.5 in the cloud droplet size distribution, which is an update from the 5.0 used by Gordon et al. (2020). Condensation of water vapor onto cloud droplets is represented with the "saturation adjustment" assumption, which means supersaturation is not prognostic and droplets are assumed to be in equilibrium at the end of each model time step. In the regional model, the onset of non-zero fractional cloud cover starts when the grid mean relative humidity exceeds 96% to account for sub-grid variability, as discussed in Boutle et al. (2016). The coupling of the CASIM scheme to the sub-grid cloud fraction scheme is discussed in detail by Field et al. (2023). Liquid water content is passed from the $4\,km$-resolution model to the $500\,m$ model through the boundaries, while the $N_d$ at the boundaries is kept constant at $100\,cm^{-3}$. The coupling of CASIM to the GLOMAP aerosol scheme is described by Gordon et al. (2020). Aerosol number concentration in the soluble Aitken, accumulation and coarse modes, and the diameters and volume-weighted hygroscopicities of these modes are passed from the aerosol code to the activation scheme in the cloud microphysics code.

In all configurations of our model, the ARG activation parameterization (Abdul-Razzak and Ghan, 2000), is used to activate aerosols at cloud base or when new cloud forms. This scheme was also used successfully for fog by Jia et al. (2019) and Yan et al. (2021) in WRF-chem and by Poku et al. (2019) with the CASIM microphysics scheme in a large eddy simulation. The ARG parameterization calculates $N_d$ based on the aerosol size distribution and on the change in ambient supersaturation, which relies only on updraft speeds to generate cooling in this paper (radiative cooling will be introduced in the companion paper). Above cloud base in existing clouds, the supersaturation is determined from a balance assuming a steady-state between a source (adiabatic cooling due to ascent) and the sink to existing cloud droplets (Gordon et al., 2020). Since there is a sub-grid cloud fraction scheme, both this approach and the ARG activation parameterization can be active in the same grid box.

Aerosol activation is simulated whenever there is a tendency for water vapor to condense rather than evaporate. At each timestep, if the activation scheme is called, it updates the $N_d$ if the $N_d$ calculated at that timestep exceeds the $N_d$ that existed in the gridbox before the activation scheme was run. We believe this procedure was first introduced by Clark (1974) and is commonly used in regional and global weather and climate models (e.g. Lohmann, 2002). As a result, we expect that $N_d$





depends on the highest updraft speed the model simulates since the formation of the cloud or fog in an air parcel, as well as other processes such as sedimentation and advection.

The default model configuration uses a minimum updraft speed of $0.01\,\mathrm{ms}^{-1}$ in the activation scheme (so any updraft speeds
lower than this are set to $0.01\,\mathrm{ms}^{-1}$ for activation when the criteria described in the next paragraph are met). In this study, most of our simulations use the resolved, grid-average, updraft speed in the ARG activation scheme, with no representation of sub-grid turbulence. This practice is frequently used in other cloud-resolving models (for horizontal grid resolutions at or below 1 km) such as the Regional Atmospheric Modeling System, RAMS (Saleeby and Cotton, 2004), and was used in the paper describing CASIM in the UM Field et al. (2023). However, sub-grid turbulence is likely an important source of updrafts
for activation of aerosols in fog. Despite earlier work (Malavelle et al., 2014; Gordon et al., 2020) we do not yet have a robust approach to represent sub-grid-scale updraft speeds across all cloud types, including fog, and across varying model grid resolutions. Addressing this limitation fully requires a dedicated, comprehensive study. Nevertheless, in this paper we conduct two additional simulations to understand the possible effects on simulated $N_d$ of including a realistic contribution from sub-grid-scale turbulence to updrafts in activation.

In the CASIM microphysics as described by Field et al. (2023), the activation scheme is called when the LWC in a gridbox increases, but droplets deactivate only if cloud fraction decreases, irrespective of LWC. For consistency, we instead choose to call the activation scheme when the cloud fraction remains the same (for example in 100% foggy gridboxes) or increases, and is not dependent on a change in the LWC. We include this change in all of our simulations. Usually, changes in LWC correlate with changes in cloud fraction, so we expect this change to have minimal effects on $N_d$.

Aerosols are not removed or advected separately when they activate to droplets. Hence, in-cloud processing of aerosols is not represented, although sulfate mass does increase via aqueous chemical reactions, depending on LWC and concentrations of sulfur dioxide, hydrogen peroxide, and ozone. Aerosols are only removed irreversibly during significant rain or snow, depending on the autoconversion and accretion rates through which cloud droplets are converted to precipitation, as described by Mulcahy et al. (2020). The rain and snow rates are also used to determine the rate of impaction scavenging of aerosol
by precipitation. In our simulations, therefore, aerosols in fog droplets are only removed via dry deposition; including an enhancement to aerosol removal due to the faster sedimentation rates of droplets compared to aerosol particles could be a useful future development.

## 4 Testing improvements to simulated activation due to adiabatic cooling

### 4.1 Default ARG Scheme (Def-ARG)

In our default simulation, termed Def-ARG hereafter, we use the ARG scheme with in-cloud activation (as described earlier) to examine whether the parameterization developed for the clouds can simulate fog realistically.





## 4.2 ARG scheme with adjusted parameters (Mod-ARG)

The semi-empirical ARG scheme for aerosol activation was not, as far as we know, designed for the low updrafts observed in fog (Abdul-Razzak and Ghan, 2000; Ghan et al., 2011), where the effect of the kinetic limit on the rate of water uptake during
activation can often become important, especially in polluted conditions. These kinetic limitations are not accounted for in the ARG parameterization (Nenes et al., 2001; Phinney et al., 2003), though they are in some more sophisticated alternatives (e.g. Morales Betancourt and Nenes, 2014). Furthermore, when comparing predictions of the ARG parameterization to the Pyrcel cloud parcel model of Rothenberg and Wang (2016), we found it exhibits significant biases for the geometric standard deviation of the accumulation mode used in GLOMAP, 1.4, which is smaller than the value used in most other modal aerosol
microphysics schemes.

In the ARG activation scheme, activated $N_d$ are calculated from maximum supersaturation $S_{max}$. The expression for $S_{max}$ is the following:

$$S_{max} = \left\{ \sum_i \frac{1}{S_{mi}^2} \left[ f_i \left( \frac{\zeta}{\eta_i} \right)^p + g_i \left( \frac{S_{mi}^2}{\eta_i + 3\zeta} \right)^{3/4} \right] \right\}^{-\frac{1}{2}} \tag{1}$$

where $i$ indexes aerosol modes in a multi-modal lognormal size distribution and $p = 3/2$.

The empirical parameters $f$ and $g$, are functions of aerosol mode width $\sigma$. $S_{max}$ depends on the critical supersaturation $S_m$ of a particular mode and two non-dimensional parameters $\eta$ and $\zeta$ which are functions of updraft speed, surface tension and thermodynamic parameters, and are independent of the widths of the size modes. Following Equation 6 in Abdul-Razzak and Ghan (2000), for each aerosol mode $f$ and $g$ were determined by comparison to a parcel model as:

$$f_i = 0.5 \exp\left(2.5 \ln^2 \sigma_i\right)$$
$$g_i = 1 + 0.25 \ln \sigma_i \tag{2}$$

Lower values of $f$ and $g$ would lead to higher activation fraction (higher $N_d$). In a separate study (Ghosh et al., 2024b), we determined improved values of $f$ and $g$ by comparing the parameterization to the Pyrcel model simulations. We also determined a modified value of $p$ which substantially improves the performance of the scheme in cases where kinetic limitations to droplet activations are critical. We use the following modified equations to reduce biases in $N_d$ predicted by the parameterization:

$$f = 0.0135 \, e^{2.367 \sigma_{acc}}$$
$$g = 1.1058 - 0.315 \, \sigma_{acc}$$
$$p = -0.5073 + 1.5088 \, \sigma_{acc} - 0.3699 \, \sigma_{acc}^2 \tag{3}$$

In Table 2, we list the parameters used in the Def-ARG and Mod-ARG simulations; for Mod-ARG, these are calculated from this Equation 3.



**Table 2.** Different values of $f$, $g$ and $p$ for for Def-ARG and Mod-ARG. In Def-ARG, different values of $(f, g)$ are used for Aitken, accumulation and coarse modes. In Mod-ARG, $f$ and $g$ are constant for all aerosol modes following Ghosh et al. (2024b). $acc$ refer to accumulation mode aerosols. Other aerosol modes make unimportant contributions to activation in our simulations.

| Simulation | $f_{acc}$ | $g_{acc}$ | $p$ |
|---|---|---|---|
| Def-ARG (accumulation mode) | 0.66 | 1.08 | 1.50 |
| Mod-ARG (all modes) | 0.37 | 0.67 | 0.88 (for $\zeta/\eta_i > 1$) |
| | | | 1.50 (for $\zeta/\eta_i < 1$) |

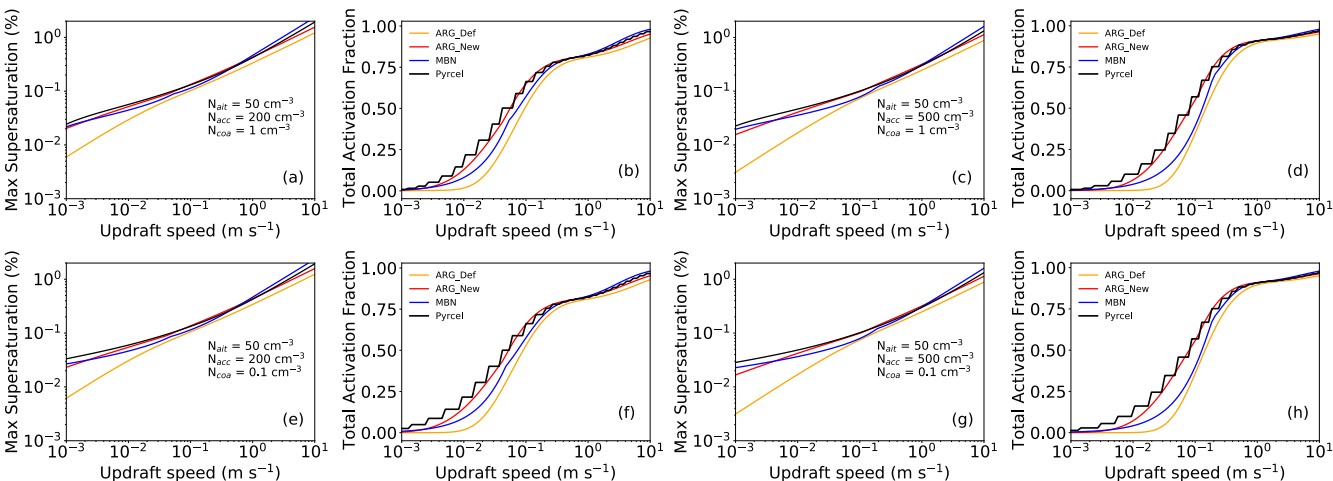

**Figure 2.** Subfigures (a, c, e, g) show maximum supersaturation, while subfigures (b, d, f, h) show total activation fraction as a function of updraft speeds for the Pyrcel model, default ARG, modified ARG, and the Morales Betancourt and Nenes (2014) (MBN) schemes. Each pair of subfigures (a,b; c,d; e,f; g,h) represents a different aerosol size distribution. The diameters of the Aitken, accumulation, and coarse mode aerosols are fixed at 40 nm, 200 nm, and 750 nm, respectively, with a hygroscopicity of 0.3 for all modes. The temperature is set at 283K and pressure at 1 atm for all cases.

In Figure 2, we compare the performance of the modified ARG scheme when run offline, to Pyrcel simulations in conditions representative of fog, with updraft speeds ranging from $10^{-3} - 10^{-1} \, \mathrm{ms}^{-1}$. We also compare the performance of the ARG parameterization with the Morales Betancourt and Nenes (2014) activation parameterization (hereafter MBN), which is more physically accurate but computationally expensive compared to the ARG scheme. We define the activation fraction as the number of droplets predicted by Pyrcel or the parameterization divided by the sum of the aerosol number concentrations

in the Aitken, accumulation and coarse modes. The Aitken mode here corresponds only to GLOMAP's Aitken soluble mode; the Aitken insoluble mode typically has higher aerosol concentrations within it but these do not activate. The number concentrations of the accumulation and coarse aerosol modes are varied across the subfigures (the Aitken number concentrations are kept fixed) at 283 K temperature, and 1 atm pressure, representative values for ParisFog. In the parcel simulations, the geometric standard deviations correspond to those set in GLOMAP at 1.59, 1.4 and 2.0 for Aitken soluble, accumulation and





coarse aerosol modes. The hygroscopicity of all modes is fixed at 0.3. We find that the default ARG scheme (yellow lines) always significantly underestimates the activation fraction. The modified scheme (red lines) performs much better than the default ARG scheme and often better than the MBN scheme.

For example, at updraft speeds between 0.01 and 0.1 ms$^{-1}$, the Pyrcel model usually activates 20- 60% aerosols, while the default ARG parameterization activates 0-50%. In comparison, the updated ARG parameterization activates similar fractions

to Pyrcel. In our UM simulations of fog, cases where the default scheme performs poorly (such as those in Figure 2), occur frequently. Thus, in simulation Mod-ARG, we include our modifications to the ARG parameters (Table 2) to improve the performance of the ARG scheme. In Section 5, we show that this modification substantially improves the model performance.

### 4.3 Modified ARG Scheme with Updated Hygroscopicities (Mod-Kappa)

Aerosol activation in the model depends upon volume-weighted hygroscopicities for the aerosol modes, described using the

Kappa-Kohler approach of Petters and Kreidenweis (2007). However, these numbers are not state-of-the-art in the standard configurations of the climate model (HadGEM3-GC3.1 and UKESM1) used in the most recent Coupled Model Intercomparison Project simulations (Mulcahy et al., 2020, 2023) nor in the NUMAC setup described by Gordon et al. (2023). By default, sulfate (and nitrate) are assumed to fully dissociate in solution, leading to a kappa value for sulfate, for example, of 0.97, while the kappa value of organic carbon is zero. A large corpus of literature provides convincing evidence (e.g. Petters and Kreidenweis,

2007; Schmale et al., 2018) that organic carbon is slightly hygroscopic and the hygroscopicity of sulfate aerosol is lower than 0.97, because it does not fully dissociate in solution.

In this simulation (hereafter termed as Mod-Kappa) we assign a $\kappa$ of 0.1 to OC, and, via an approximate approach, we arrive at kappa values close to the recommended 0.61 for ammonium sulfate and bisulfate and 0.73 for sulfuric acid (Schmale et al., 2018; Fanourgakis et al., 2019) and 0.67 for ammonium nitrate (Petters and Kreidenweis, 2007). This is achieved by setting

the kappa values of the sulfate component in our model to 0.73, nitrate to 0.83, and ammonium to zero. This solution maintains the simplicity of having a single kappa value for each component species (e.g. sulfate) that does not vary according to which other species are present.

In Table 3 we list the hygroscopicities for different aerosol species used in Mod-ARG and Mod-Kappa. When we run our simulation with these new kappa values, we also include changes to the ARG scheme from the Mod-ARG simulation.

**Table 3.** Different values of hygroscopicities for different aerosol components used in the Mod-ARG and Mod-Kappa simulations. Nitrate and ammonium were not included in climate simulations for CMIP6.

| Simulation | BC | OC | Sulfate | Sea Salt | Nitrate | Ammonium |
|------------|-----|-----|---------|----------|---------|----------|
| Mod-ARG | 0.0 | 0.0 | 0.97 | 1.34 | 0.87 | 3.54 |
| Mod-Kappa | 0.0 | 0.1 | 0.73 | 1.50 | 0.83 | 0.0 |



# 5   Results

## 5.1   Location and Timing of Fog

We start our model evaluation by comparing the location of fog in the 4 km resolution model to satellite imagery. This evaluation demonstrates qualitatively how the model can simulate fog over a large domain with variable topography and some ocean, but the quantitative analysis in this paper focuses on the 500 m-resolution model. Our satellite dataset is the fog product from the Spinning Enhanced Visible and Infrared Imager (SEVIRI) onboard the Meteosat Second Generation (MSG) satellites, operated by the European Organisation for the Exploitation of Meteorological Satellites (EUMETSAT), Schmetz et al. (2002).

In Figure 3, we show the location of the fog from simulation Def-ARG at 4 km resolution, on about half of the days during our case study period at 03:00 UTC. The satellite images (which correspond roughly to the 4 km resolution model domain) on the top panel show the location of fog in different colors depending on fog temperature. For example, the simulated fog near the center of the domain is much thicker on 15$^{th}$ Nov than on 22$^{nd}$ Nov. The SIRTA observatory is shown by a red dot near the center of the domain. At 03:00 UTC, we have some coverage of fog at the center of the domain on all days. On 15$^{th}$ and 16$^{th}$ November, the fog is much denser in the observations but on 20$^{th}$, 22$^{nd}$, and 23$^{rd}$ November, the fog is patchy. The model could realistically simulate fog towards the English Channel and UK, although we underestimate marine fog on the 16$^{th}$ and 23$^{rd}$. Fog simulated in the top right corner towards Germany on 15$^{th}$ and 16$^{th}$ is less widespread than satellite observations.

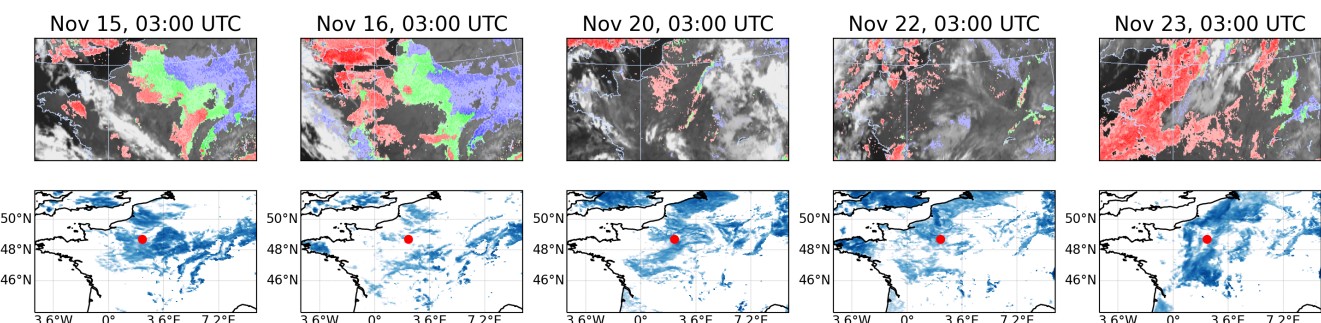

**Figure 3.** Location of the fog at 03:00 UTC during different representative fog events. The top panels show images from the Spinning Enhanced Visible and Infrared Imager (SEVIRI). Red refers to fog with temperatures $> 1°C$, blue refers to fog $< -1°C$, and green refers to fog with temperatures between $1°C$ and $-1°C$. The bottom panel shows the location of fog in the 4 km model. We show LWC (at the surface) across the domain in blue. Thicker fog is represented by a darker shade. The SIRTA observatory at the center of the domain is denoted by a red dot.

Overall, during our simulated period between 14$^{th}$ and 26$^{th}$ November 2011, 11 fog events were observed at SIRTA while, in our default simulation, 13 fog events were simulated in our 500 m-resolution domain. The onset of fog is usually well after dusk (21:00 UTC to 03:00 UTC) on most of the days. The fog layer gradually develops and finally dissipates in the morning due to shortwave heating. When the fog top altitude is lower than 18 m, which is the case for the 4 observed events from 18$^{th}$ to 22$^{nd}$, Mazoyer et al. (2019) characterized the fog as 'thin', while the 7 other cases are thick fog.



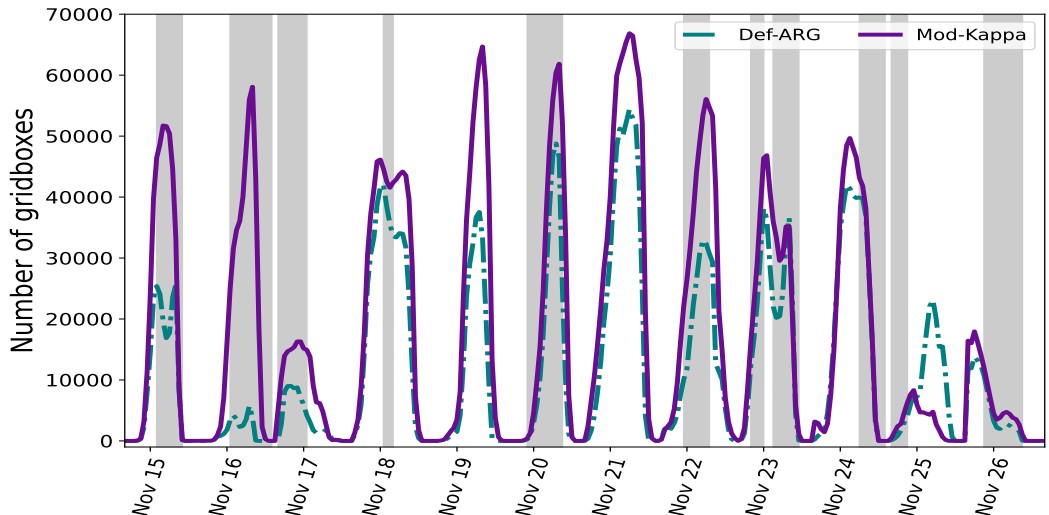

**Figure 4.** Timeseries of number of foggy gridboxes (at the surface) from the 500 m-resolution model for the two of our simulations that are substantially different (Mod-ARG and Mod-Kappa behave very similarly). If the entire domain was foggy, the total would be 67600 (260×260). Foggy periods in the observations are shown in shaded grey. Tick marks on the x axis are at midnight UTC time.

Figure 4 shows a time series of the number of foggy gridboxes at the surface in the 500 m model from simulations Def-ARG and Mod-Kappa. We do not show Mod-ARG since it performs similar to Mod-Kappa (discussed later). The visibility parameterization of Gultepe et al. (2006) suggests that if $N_d$ is 200 cm$^{-3}$ and LWC is 0.005 g m$^{-3}$, visibility would be ~ 1 km. Therefore, we define foggy gridboxes as those having at least 0.005 g m$^{-3}$ LWC (similar to Mazoyer et al. (2019)) and at least 20% cloud cover. Changing these thresholds by a factor of 2 does not make any significant difference. To avoid artefacts due to the interpolation of the grid from 4 km to 500 m resolution, we do not include the 20 gridboxes near the edges of the domain in the figure, regardless of the presence of fog. Therefore, the maximum number of foggy gridboxes at any given model level is 67600 (260×260).

During the two weeks we simulated, the model produces a fog with a realistic lifetime for most of the fog events, however, the onset and dissipation times are not always correct (fog on the morning of 18$^{th}$ Nov starts too early, for example). On the mornings of 19$^{th}$, 21$^{st}$ and 25$^{th}$ November, the model predicts widespread fog but there was no fog in the observations at SIRTA. Some discrepancies are expected considering the difficulty involved in simulating fog accurately at the correct location using NWP models without data assimilation (e.g. Bergot et al., 2005; Zhou and Ferrier, 2008; Velde et al., 2010; Bergot, 2013; Steeneveld et al., 2015; Boutle et al., 2016, 2018; Martinet et al., 2020). However, we simulate enough fog to be able to examine aerosol effects on $N_d$ and fog microphysics.





## 5.2 Evaluation of temperature and relative humidity profiles and time-series


We evaluate the ability of our model to simulate temperature and relative humidity (RH) realistically near the surface. Accurate representation of these parameters is important because too high temperature or too low relative humidity can lead to dissipation of fog, and fog often needs temperature inversions to form. Two radiosonde profiles are available each day at around 23:00 UTC and 11:00 UTC. In Figure 5, we compare the night-time radiosonde profiles with our 500 m resolution Def-ARG simulation.

Fog is not usually seen at the time of the daytime profile, but the profiles are available in Figure S1.

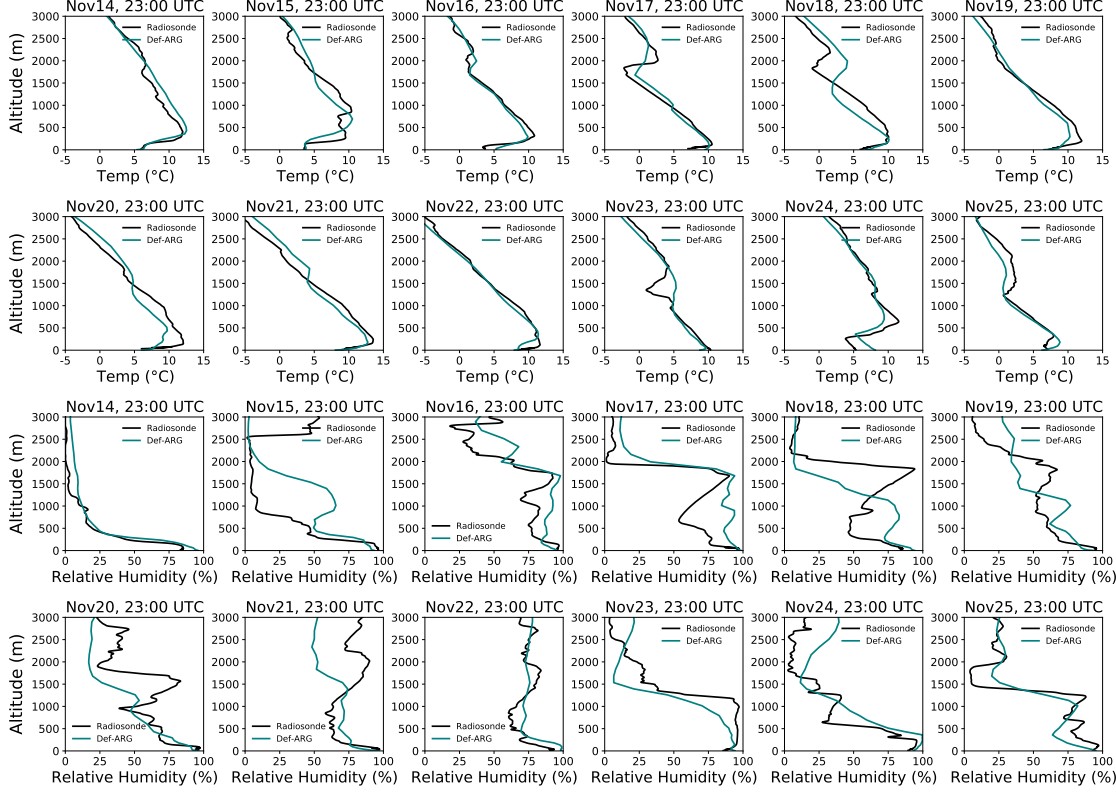

**Figure 5.** Vertical profiles of temperature and relative humidity from 500 m grid resolution model of simulation Def-ARG. The results are compared with the data from radiosondes launched from Trappes around 23:00 UTC on 14th to 25th November.

Within 200 m of the surface at night, the simulated temperature profile in the 500 m domain is usually in reasonable agreement with observations, within 1°C or so. Exceptions are on November 20th, where simulated temperatures above about 150 m altitude are approximately 2°C too low, and 24th when there is a ~2.5°C overestimation below 300 m altitude. Even an 0.5°C





bias may be enough to significantly affect the simulated fog. Nonetheless, the model can reproduce the surface inversion height
and approximate strength on all the days, except for November 20$^{th}$ when the inversion is too weak and November 24$^{th}$ when it
is too high. The higher-level inversions were not always reproduced accurately. For example, on 17$^{th}$ and 18$^{th}$, the temperature
profile above 1 km has biases of up to ~3°C.

Similar trends were observed for the RH profiles. Within 200 m of the surface, the 500 m model simulates RH realistically,
within 10% of the observation most of the times. RH near the surface has a larger positive bias on 17$^{th}$ and a negative bias on
25$^{th}$. At a higher altitude, RH has much larger biases, on most of the days. The model frequently fails to reproduce the detailed
structure. In both simulations and observations, the fog top is usually near the surface, where the model performs best, but we
must nonetheless consider that (especially) the radiative properties of the fog may be affected by biases in high cloud cover.

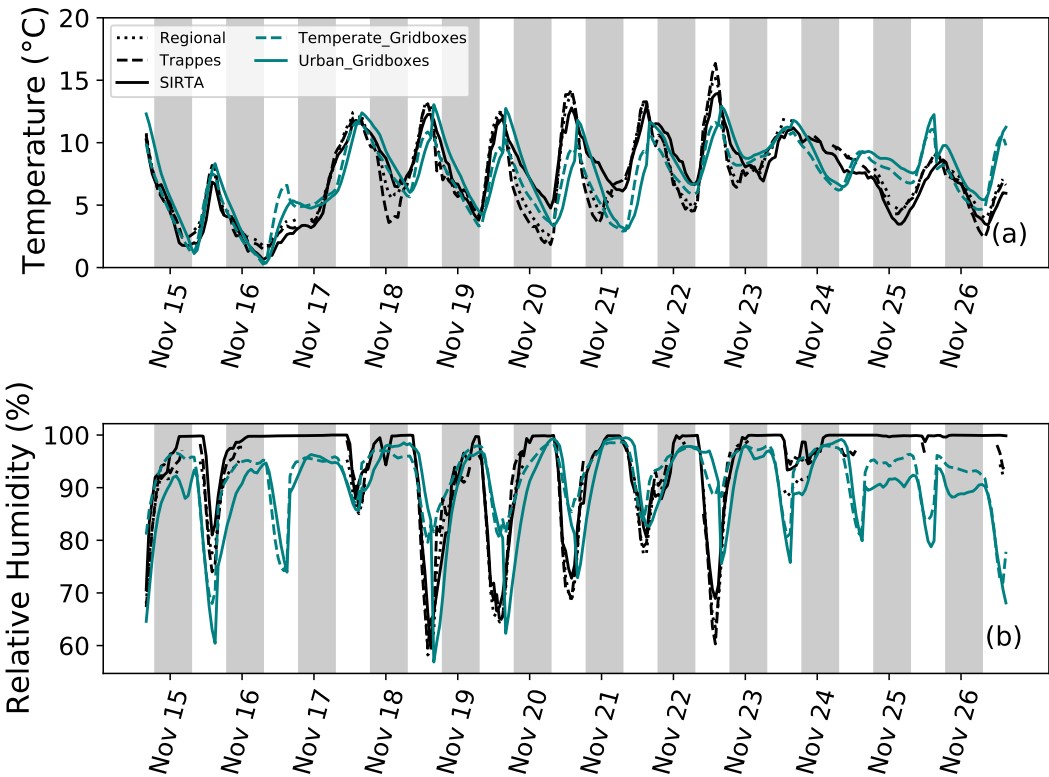

**Figure 6.** Observed and simulated near-surface temperature (a) and relative humidity (b) during different fog episodes. We show direct
measurements from the sites at SIRTA and Trappes. 'Regional' refers to weighted average temperature and RH calculated from measurements
at Orly, Montsouris and Trappes, following section 3.2.1 of Chiriaco et al. (2018). The simulated 'Temperate' and 'Urban' timeseries are
averages of all gridboxes denoted as 'Temperate Grass' and 'Urban'. Less frequently occurring surface types, such as water, are not shown
here. Times are UTC and the shaded region denotes nighttime.





Figure 6 compares the observed temperature and RH timeseries with simulated temperatures (a) and RH (b) at 'Temperate Grass' and 'Urban' surface types. We show direct measurements from the sites at SIRTA and Trappes. We also plot the regional

weighted average using measurements from sites at Paris-Montsouris, Orly, and Trappes following section 3.2.1 of Chiriaco et al. (2018). Nighttime is denoted by the shaded regions. Urban tiles are usually warmer and have lower RH than grass due to heat island effects. Features of the diurnal cycle which is relevant to fog for both temperature and RH are well simulated by the model.

For temperature, the R value of the timeseries is 0.80, and the normalized mean bias (NMB), defined as

$$\text{NMB} = \frac{\sum_i (M_i - O_i)}{\sum_i O_i}$$

for simulated values $M_i$ and observations $O_i$, is -0.08; for RH R is 0.49 and NMB -0.03. These statistics are calculated by comparing the mean profiles from four sites with mean profiles from model. There are missing data from observations on 17th and 24th-26th Nov from one or more sites. The model simulates the temperature timeseries within 2 degrees, except on the night of 24th-25th where a bias was already identified from the radiosonde profile. The RH is also biased low during this time,

though the relatively low R value for the RH timeseries appears to be mostly due to an overestimation of RH by the model when RH is low (as on 22nd), and when fog is not present.

### 5.3  Evaluation of Aerosols

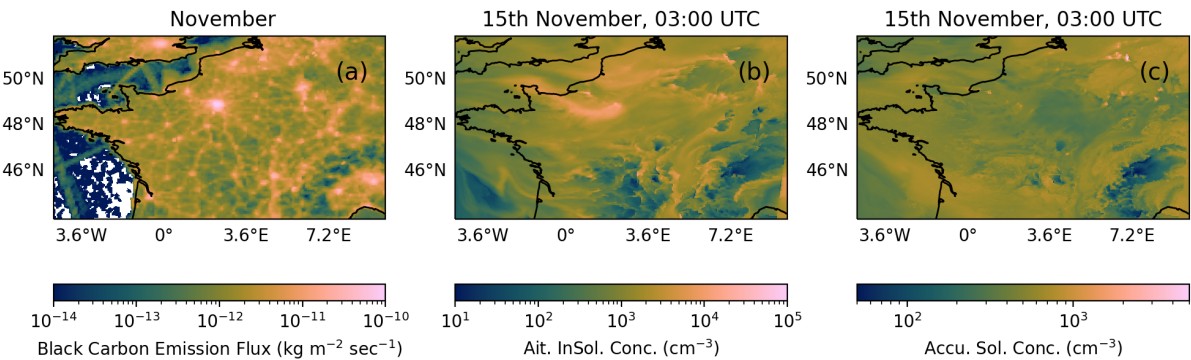

**Figure 7.** The black carbon emission flux at the surface (a), from the high resolution EDGAR-HTAP inventory, in the 4 km model for the month of November. Hotspots such as London and Paris can be identified. Subfigure (b) shows the Aitken insoluble mode number concentration and (c) shows the accumulation soluble mode particle number concentration, as simulated by the 4 km model in simulation Def-ARG near the surface at 03:00 UTC on 15th Nov.

We investigate the ability of the model to simulate aerosols realistically in simulation Def-ARG. All simulations use the same set of boundary conditions and emissions inventory, so we do not expect the aerosol concentration to differ substantially

between our simulations. Figure 7 demonstrates qualitatively how aerosols are simulated in our 4 km-resolution model. Subfigure (a) shows the black carbon (BC) emission flux, to illustrate the major emission sources in our domain while using the



high resolution EDGAR-HTAP inventory. BC particles contribute to number concentrations in the 'Aitken insoluble' mode (Subfigure (b)). Subfigure (c) shows the number concentration of accumulation 'soluble' mode particles near the surface. We infer that CCN concentrations (which are mostly activated accumulation mode aerosols) are not obviously correlated with major emissions sources on this date.

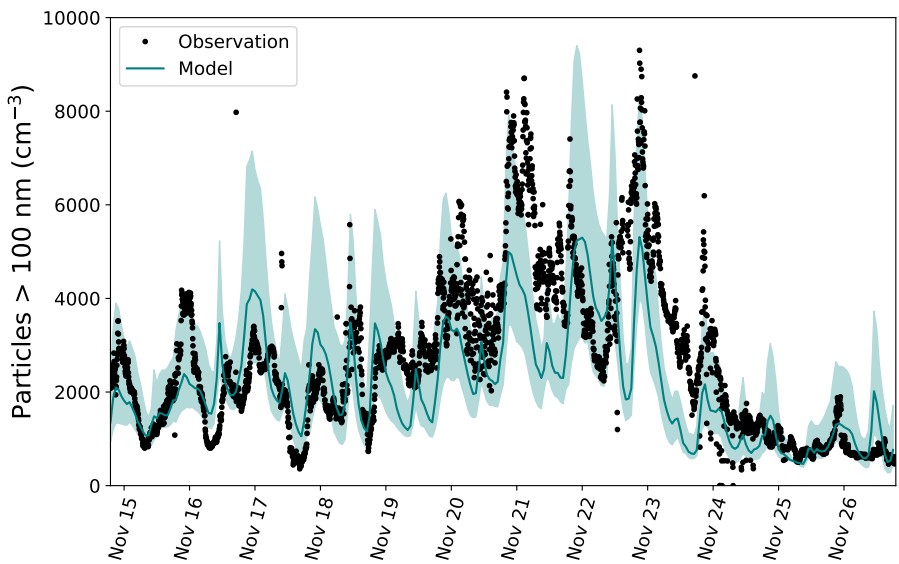

**Figure 8.** Timeseries of number concentration of aerosols greater than 100 nm in diameter as simulated by the 500 m model in the Def-ARG simulation at 5 m altitude, compared with the observations from SMPS in the same size range. The model median is represented by the green line and the shaded region represents the interquartile range. Times are UTC.

In Figure 8, we compare the timeseries of simulated number concentration of >100 nm dry diameter aerosols with the observations from the SMPS at SIRTA, near the surface. The simulated median number concentration from the 500 m resolution Def-ARG simulation is shown by a green solid line, and the shaded region shows the interquartile range across the model domain. The region within 20 gridboxes of the boundary is excluded, as for our analysis of fog. As mentioned earlier, Figure S2 shows timeseries of the number concentrations in the different GLOMAP lognormal aerosol modes. The simulated aerosol number concentration agrees reasonably well with the observations, with a correlation coefficient of 0.71 and a normalised mean bias of −0.18. These biases are typical, or better than typical, for simulated aerosol number concentrations in regional and global models (e.g. Baranizadeh et al., 2016; Williamson et al., 2019; Ranjithkumar et al., 2021).

In Figure 9, we show the percentage of different aerosol species by mass as a function of time. Crippa et al. (2013) carried out a source apportionment study over Paris during winter in November with an aerosol mass spectrometer and aethalometer. They found the following composition of aerosol by mass: 36% organic carbon (OC), 28.4% nitrate, 15.7% sulfate, 11.9% ammonium , 6.8% black carbon (BC), and 1.2% chloride. Another study in northern France in 2015-6 (Roig Rodelas et al., 2019), found similar aerosol composition. On average, our simulations have 28.2% OC, 32.7% nitrate, 9.4% sulfate, 12.3%



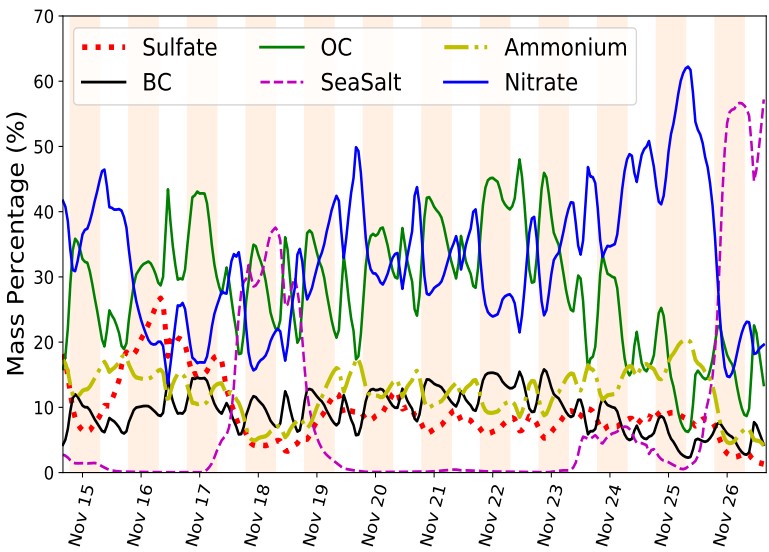

**Figure 9.** Percentage of different aerosol species by mass as a timeseries (UTC time) throughout the simulation period. The mean values from the gridboxes in the 500 m resolution Def-ARG simulation near the surface are shown. Nighttime is denoted by the shaded regions. All species the model represents with the GLOMAP microphysics scheme are shown; dust cannot participate in CCN activation.

ammonium, 9.4% BC, and 8.0% sea-salt (4.8% chloride) aerosol mass (see Table 4). Hence, most likely our simulations are
realistic, but they probably underestimate organic aerosol. Secondary organic aerosol (SOA) formation from anthropogenic volatile organic compounds and isoprene is not included in our model and could explain the bias.

**Table 4.** Percentage composition of aerosols over Paris from Crippa et al. (2013) and the 500 m model (simulation Def-ARG, at the surface) is listed here.

|                      | BC (%) | OC (%) | Sulfate (%) | Chloride (%) | Nitrate (%) | Ammonium (%) |
|----------------------|--------|--------|-------------|--------------|-------------|--------------|
| Crippa et al. (2013) | 6.8    | 36     | 16          | 1.2          | 28          | 16           |
| 500 m Model          | 9.4    | 28     | 9.4         | 4.8          | 33          | 12           |

Figure 10 shows the aerosol dry and ambient or hydrated size distributions from our 500 m Mod-Kappa simulation, compared to observations at 03:00 UTC on 15[th] and 16[th] November. The solid red line represents the total number of particles (and also $N_d$ for the ambient-humidity size distribution subfigure) simulated by the model. In dashed pink line we show the $N_d$ distribution
from Def-ARG simulation. Similar plots at the same time on 20[th] and 26[th] November are shown in Figure S3.

On the days shown in Figure 10 and Figure S3, the model slightly overpredicts the aerosol concentrations in the Aitken insoluble mode. The aerosol size distributions are better in agreement for the accumulation mode, except on 20[th] where it is underestimated. The ambient-humidity size distributions show that a more substantial underestimate of the number concentra-





**Figure 10.** Dry and ambient aerosol size distribution on 15$^{th}$ and 16$^{th}$ November at 03:00 UTC from observations and our Mod-Kappa simulation. Only foggy grid cells are shown. The dry size distribution is measured by the SMPS only, while the WELAS and fog monitor (FM) observed the size distribution at ambient relative humidity. For this ambient size distribution, we also converted dry particle concentration data from the SMPS to ambient using $\kappa$-Köhler theory assuming a kappa value of 0.1 (and 0.3) and 100% relative humidity. The red line shows simulated total dry and ambient-humidity particle size distributions, while the dashed lines show the different aerosol and droplet modes. The dashed pink line shows $N_d$ from simulation Def-ARG.

tions is usually observed in the coarse mode ($0.01 - 1\ \mathrm{cm^{-3}}$). This seems to be the main symptom of the inability of the aerosol
microphysics scheme to accurately simulate haze, discussed earlier.

On 15$^{th}$ Nov, $N_d$ in the default simulation agrees well with the observations, but on the other three days $N_d$ is severely underestimated. However, a significant improvement is observed in Mod-Kappa, which we show in later figures is due to the change between Def-ARG and Mod-ARG. Also, so far, the difficulties in simulating haze do not appear to prevent a good




simulation of droplet concentrations. The next sections investigate in greater detail the model performance in simulating fog

droplets.

## 5.4 Evaluation of fog droplet concentration and liquid water content

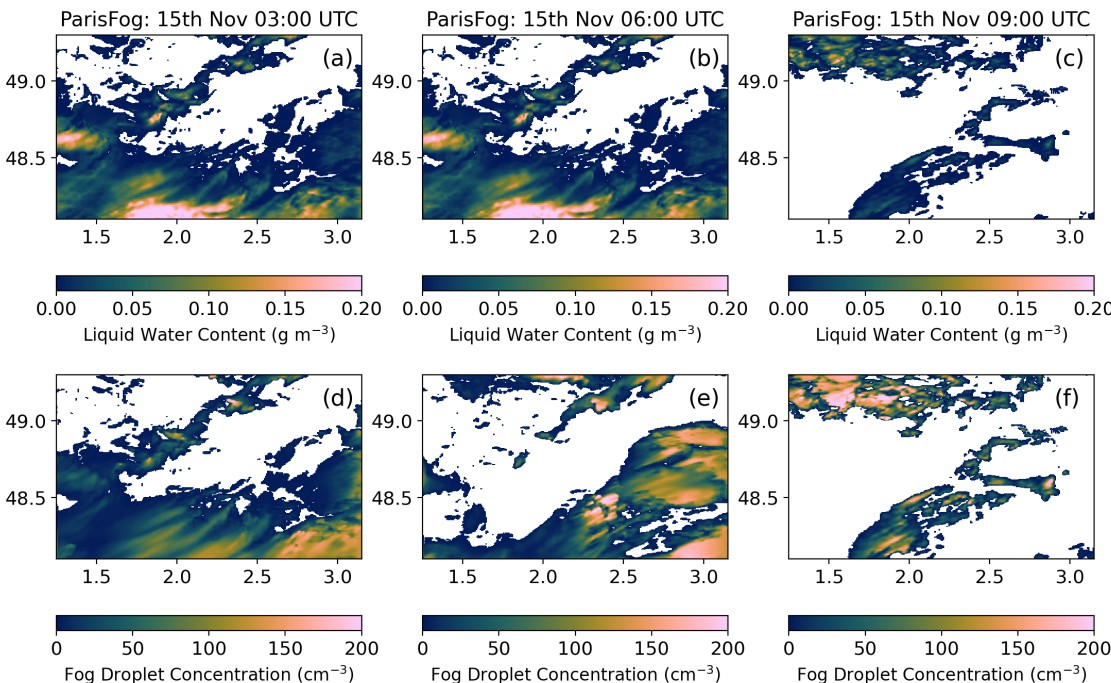

**Figure 11.** Spatial distribution of in-fog LWC (top panel) and fog $N_d$ (bottom panel) near the surface (5 m altitude) at different UTC times on 15[th] Nov from simulation Def-ARG. These results are from the 500 m resolution model. The criteria for identifying a grid box as foggy are explained in Section 5.1 and Table 5; grid cells that do not meet the criteria are white.

The 500 m grid resolution simulations show widespread but inhomogeneous fog LWC and $N_d$ during most fog events. To illustrate spatial variability, example patterns are shown from the Def-ARG simulation three times during the fog event on 15[th] November in Figure 11 at 5 m altitude. The domain is foggy at late night and early in the morning and dissipates around

09:00 UTC. Similar figures during four other fog events are shown in Figures S4, S5, S6 and S7.

In Figure 12, we plot the timeseries of in-fog $N_d$ for different fog events from our 500 m model, compared to observations from the fog monitor at SIRTA. In each grid box, in-fog $N_d$ is calculated by dividing the grid-averaged $N_d$ by the fractional cloud cover. To allow a comparison even when simulated and observed fogs are not exactly co-located as in Figure 11, we



show the medians as solid (and dashed) lines and the interquartile ranges over all of the foggy gridboxes as shaded regions.
We also impose a minimum threshold of 1000 foggy gridboxes while calculating the domain median and IQR, as shown in Table 5. This threshold corresponds to about 1.5% of the area of the entire domain and is designed to eliminate cases where fog forms only in isolated locations that are unlikely to be representative of the SIRTA observatory.

**Table 5.** Summary of different thresholds used to prepare $N_d$ and LWC timeseries plots. Min_Fog denotes the threshold number of foggy gridboxes, below which a spatial average of the model domain for a timeseries is not calculated.

| Figure | Cloud Fraction | LWC | Min_Fog |
|---|---|---|---|
| 4, S2 | 20 % | 0.005 g m$^{-3}$ | NA |
| 12, 13 | 20 % | 0.005 g m$^{-3}$ | 1000 |
| S7 | 20 % | 0.005 g m$^{-3}$ | 10 |

Simulation Def-ARG was designed to understand the performance of the 'ARG' activation scheme as implemented in the CASIM microphysics code in simulating $N_d$ in fog. The observed $N_d$ varies between $0 - 300\,\text{cm}^{-3}$ among different fog events
(except for a couple of hours on 20$^{\text{th}}$ Nov morning and 22$^{\text{nd}}$ Nov evening). We calculate the Normalized Mean Bias Factor (NMBF) and the Normalized Mean Error Factor (NMEF), defined as

$$\text{NMBF}(\overline{M} \geq \overline{O}) = \frac{\sum_i (M_i - O_i)}{\sum_i O_i}$$
$$\text{NMBF}(\overline{M} < \overline{O}) = \frac{\sum_i (M_i - O_i)}{\sum_i M_i}$$

$$\text{NMEF}(\overline{M} \geq \overline{O}) = \frac{\sum_i |M_i - O_i|}{\sum_i O_i}$$
$$\text{NMEF}(\overline{M} < \overline{O}) = \frac{\sum_i |M_i - O_i|}{\sum_i M_i}$$

Here, $M_i$ are the model data, $O_i$ is the observation data, $\overline{M}$ is the model mean, and $\overline{O}$ is the observation mean. NMBF has a range of $-\infty$ to $+\infty$, and NMEF has a range of $0$ to $+\infty$. We report NMBF and NMEF for all fog cases in Supplement Tables S1 and S2 and use them as a tool to compare the model performance among different simulations.

By comparing the median time series shown by the dashed blue line, for some of the fog cases the default simulation is in
satisfactory agreement with the observations, such as the fog cases on 15$^{\text{th}}$ and 24$^{\text{th}}$ Nov early in the morning (Figure 12a, i). For these cases, we find NMBF values of 0.72 and 0.07, and NMEF values of 0.99 and 0.72. However, most of the time the simulation Def-ARG severely underpredicts $N_d$. For example, in both fog events on 16$^{\text{th}}$, and in cases on 21 and 25$^{\text{th}}$ (Figure 12b, c, f, k), the simulated droplet concentrations are lower than observations by more than a factor of 4 (NMBFs equal to -9.0, -3.3, -8.8 and -3.8). Figure S8 shows the time series of droplet concentrations during fog events from only foggy
gridboxes within a 20×20 cell box around SIRTA. Selecting only these gridboxes does not change the significant underestimate of $N_d$ on average.

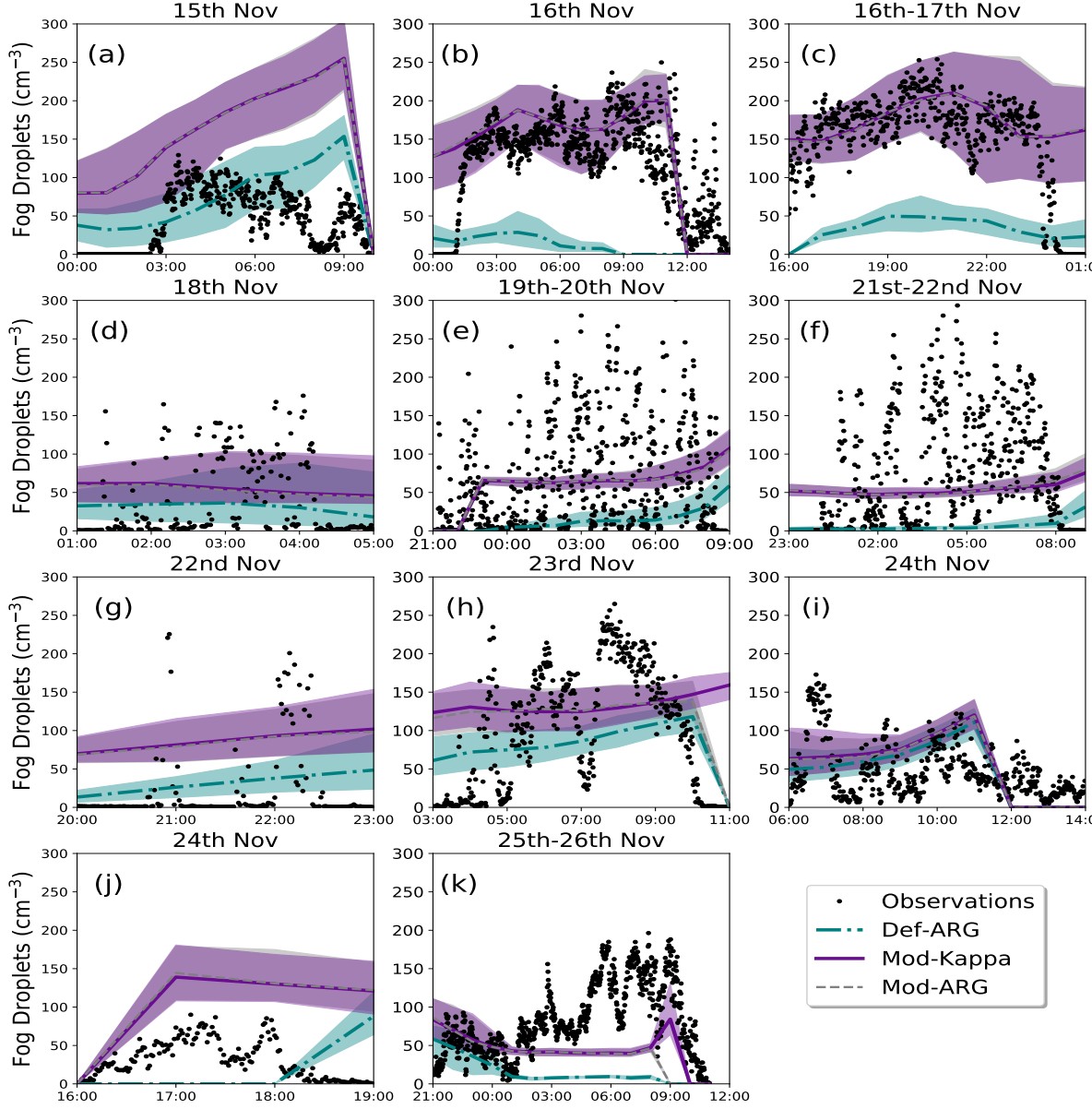

**Figure 12.** Variation of simulated and observed $N_d$ as a function of time for different fog events. 500 m model results at 5 m altitude from simulations Def-ARG, Mod-ARG, and Mod-Kappa are compared with observations at the SIRTA observatory (UTC time). Different lines represent the median value and shaded regions represent the interquartile range over the foggy gridboxes. The simulated $N_d$ in Mod-Kappa closely resembles that in Mod-ARG; they can be distinguished, for example, at 09:00 UTC in subfigure k. Figure S9 shows that including a contribution from sub-grid scale updraft velocities in the activation scheme has a similar effect as changing from Def-ARG to Mod-ARG, and that with higher updraft speeds the effect of the change from Def-ARG to Mod-ARG on $N_d$ would often be much smaller.



In the Mod-ARG simulation, we introduced improved parameters for the ARG scheme (Ghosh et al., 2024b). In the Mod-Kappa simulation, we adjusted the hygroscopicities to more realistic values, reducing that of sulfate while increasing it for organic aerosols. As expected from the activation parameterization tests shown in Figure 2, in simulation Mod-ARG $N_d$
usually increases significantly compared to Def-ARG. Model performance is improved in most fog events. During thick fog events on the 16[th] (subfigures b, c), NMBF (NMEF) changed from -9.0, -3.3 (9.2, 3.5) in Def-ARG to 0.23, 0.24 (0.37, 0.31) in Mod-ARG. In Mod-ARG, simulated $N_d$ is around 150-200 cm$^{-3}$ (similar to observations) compared to 0-50 cm$^{-3}$ ( a factor of 10 underestimation) in Def-ARG. Similar performance improvements are observed on 19[th] and 21[st] November fog cases. The NMBF (NMEF) improves from -5.2, -8.8 (5.9, 9.7) to -0.42, -0.18 (1.01, 0.85). In 18[th] and 24[th] (morning) fog cases
(subfigure d, i), the performance between default and updated simulations is very similar. On 26[th] November, the factor of 10 underestimation after midnight, is now within factor of 3. Overall NMBF (NMEF) changes from -3.8 (4.2) to -0.92 (0.97). On 23rd and on this day, the fog dissipation is better simulated in Mod-Kappa compared to Mod-ARG. For the 22[nd] November evening fog event (Subfgure g), on average the default simulation is better, but relatively higher $N_d$ between 22:00 and 23:00 UTC, is slightly better simulated in Mod-ARG, On 24[th] November evening (subfigure j), the default simulation fails to produce
fog most of the time, while a good agreement of $N_d$ is simulated with our improvements during fog formation and growth. In both simulations, the dissipation of fog is not accurate in the model. On 15[th] Novemeber, $N_d$ is overestimated (NMBF increases to 2.7 from 0.72 in the default simulation) as a result of our changes; however, the overestimation is slightly higher than a factor of 2.

We did not observe significant differences in droplet concentration ($N_d$) between the Mod-ARG (gray in Figure 12) and
Mod-Kappa (purple) simulations. This lack of difference could be due to the opposing effects of changes in sulfate, nitrate, and organic aerosols, which may lead to cancellation. However, we also do not expect strong sensitivity to hygroscopicity: Section 4.2 and Figure 6(c) of Mazoyer et al. (2019) demonstrates that, for the ParisFog cases examined in our study, observations do not suggest there is significant correlation between $N_d$ and $\kappa$.

We expect that the underestimate in $N_d$ in Def-ARG is caused by an underprediction of the fraction of aerosols activated
at low updraft speeds by the default ARG parameterization, and by an underestimation of the updraft speed in the activation scheme that results from not resolving all the turbulence. Figure S9 shows the diagnosed sub-grid vertical velocity in our simulations (subfigures c and f). Qualitatively, the values seem likely to be realistic if compared with the turbulent kinetic energy (TKE) shown in ParisFog observations by Stolaki et al. (2015) on 15 November, though the comparison is imprecise since TKE contains horizontal contributions, and some vertical velocity variance is resolved by our model and therefore not
included in the sub-grid component.

In Figure S10, we show two simulations in which the updraft speeds in Def-ARG and Mod-ARG are increased by including a component to represent sub-grid-scale, unresolved vertical turbulence. This turbulence component includes an ad-hoc tuning factor of 0.2 to keep $N_d$ realistic; as mentioned earlier, dedicated future studies are needed to fully explore the turbulence contribution. Figure S10 confirms that the underestimation of $N_d$ in Def-ARG could be largely resolved either by using Mod-
ARG parameters or by adding a sub-grid component of updraft speeds with a tuning factor. Including both modifications leads to an overestimate of $N_d$, with NMB positive for all fog events and above 1.0 (a factor 2 overestimate) in 6 out of 11. In the



companion paper, we will explore strategies to mitigate the overestimations of $N_d$, which may also be beneficial when the impacts of sub-grid turbulence are more fully addressed in future. Furthermore, Figure S10 shows that the differences in $N_d$ between Def-ARG and Mod-ARG in Figure 12 would be substantially smaller if we included a subgrid turbulence contribution to updrafts for aerosol activation, a result that could also be inferred from Figure 1.

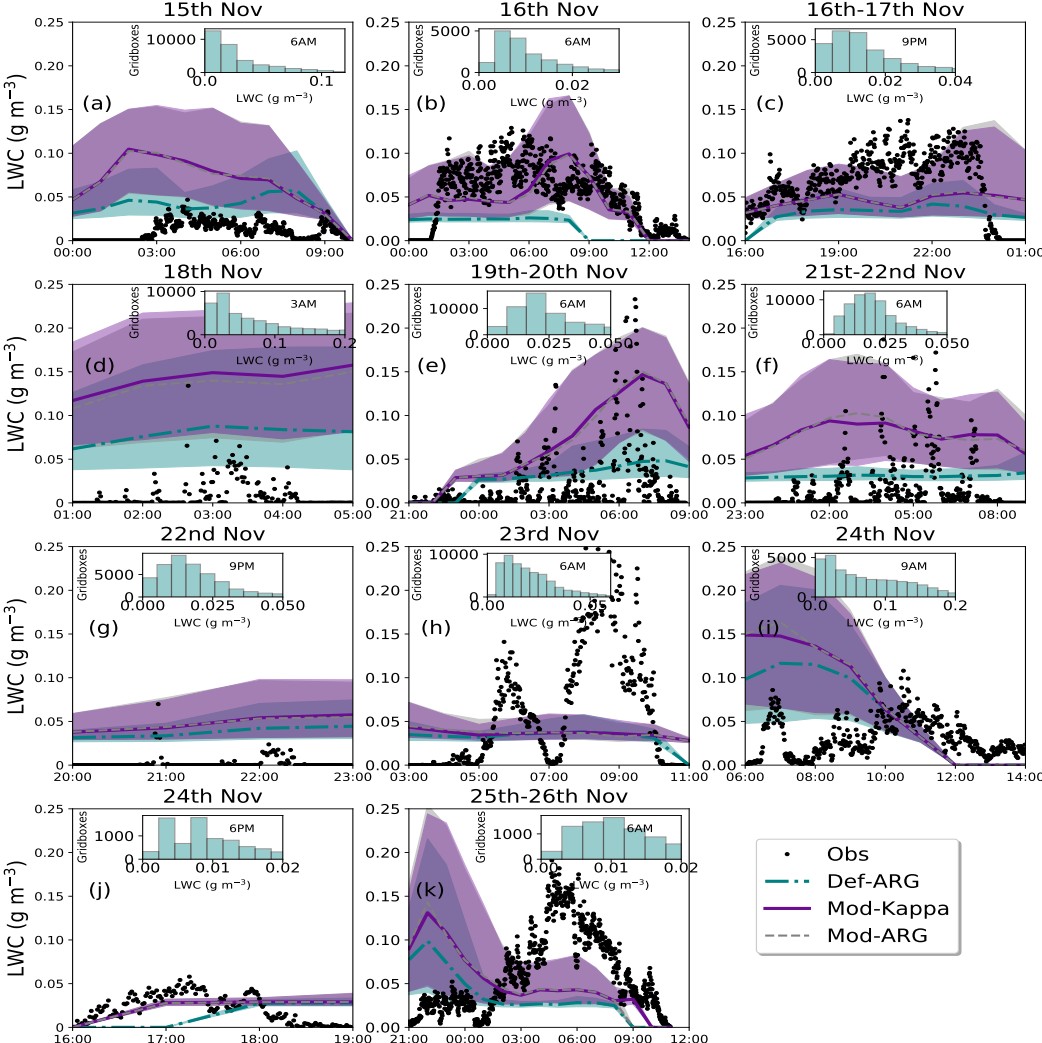

**Figure 13.** Variation of LWC as a function of time (UTC) for different fog events in the 500 m resolution regional model. Subfigures (a-k) compare all the simulations with the observations for the different fog cases. The solid lines and shaded regions represent the median and interquartile range from foggy gridboxes. The inset plots show histogram of in-fog liquid water content at different times of the fog events. The minimum LWC visible in the plots of around 0.025 g m⁻³ is due to the thresholds for defining gridboxes as foggy, as described in the text.





In Figure 13, we compare the timeseries of in-fog LWC in foggy gridboxes to observations. Subfigures (a-k) show different fog cases. In-fog LWC is calculated by dividing the grid average LWC by the cloud cover and the same gridbox number threshold as in previous figures is applied. As shown in Figure 4, most of the time the model simulated enough foggy gridboxes in the domain during all the events for the statistical properties (median and interquartile range) to be meaningful. At other

times, the LWC is zero. When there are sufficient foggy gridboxes, the LWC and cloud fraction thresholds for defining these gridboxes as foggy leads to a minimum in the LWC of around $0.025\,\mathrm{gm^{-3}}$. Since low LWC grid boxes also tend to have low cloud fraction, the minimum here is the LWC threshold of $0.005\,\mathrm{gm^{-3}}$ divided by the cloud fraction threshold of 20%. The interquartile ranges shaded on the timeseries in Figure 13 show the general nature of the variability between gridboxes, while the inset figures on each subplot are histograms of in-fog LWC that give more detail at a particular time of the corresponding

fog event, from the default simulation. In all fog events irrespective of thickness and type, there is substantial spatial variability of the simulated LWC in the 500 m resolution model as shown in the inset histograms. Therefore the observations at the SIRTA observatory may be quite different from the domain averages we present from the model. Figure S11 shows only the $20 \times 20$ gridboxes around SIRTA; however the general trends in LWC in this figure do not look much different to Figure 13.

While our simulations are mostly able to produce fog at the right time and despite reasonable simulations of $N_d$, neither Def-

ARG nor Mod-ARG simulations have much skill in representing LWC on many of the days. Visually, trends in the timeseries only roughly match observations during three fog events of 11 (the two events on 16th and the event on 24th). This low skill may be partly an artefact of comparing a point observation with a domain average (Schutgens et al., 2017). Low skill is also expected from recent intercomparison studies (Boutle et al., 2022), in which LES and single column models were also found to have little skill in simulating liquid water path, even for a case our model is able to simulate reasonably well (see supplement

Figure S12 and the companion paper). Realistic simulations of surface $N_d$, perhaps within 50% of observations, may (or may not) be a necessary condition, but are clearly not a sufficient condition, for realistic simulations of LWC. Similarly to the $N_d$ timeseries, LWC is underestimated during several fog events in simulation Def-ARG, such as for fog cases on 16th Nov, 23rd Nov, around $04:00-06:00$ UTC on 21st Nov, 24th Nov, and after fog onset on 25th Nov. During these periods, the bias is larger than a factor of 5. LWC in the observations is mostly between $0-0.20\,\mathrm{gm^{-3}}$, but LWC in the default simulation is

always $< 0.10\,\mathrm{gm^{-3}}$. However, on some days, such as 15th and 20th Nov (on average), the default model performance was satisfactory (within a factor of 2 bias). Simulation Def-ARG (and other simulations as well) did not reproduce the thick fog on 23rd November. The LWC time series could be affected by errors in the simulated higher-level cloud properties on this day. The model simulates substantial cloud cover above 300 m altitude, and the temperature and RH profiles between 500 m and 3000 m altitude (Figure 5) have errors of up to $3^\circ$C and 15% respectively, suggesting that these clouds may not be realistic. In

addition, simulated aerosol concentrations were much lower than observations (Figure 8).

Similar to $N_d$, there is a large difference in LWC between Def-ARG and Mod-ARG, and minimal differences between Mod-ARG and Mod-Kappa. The large difference between Def-ARG and Mod-ARG suggests that aerosol-fog interactions are important, at least in our simulations. Changes in the ARG scheme, increasing $N_d$, result in a significant increase in LWC compared to the default simulation. However, comparing to the observations, it is unclear whether biases increase or

reduce with this change. The most obvious mechanism that could explain the sensitivity of the simulated LWC to $N_d$ in these





simulations is the size dependence of droplet sedimentation rates. Figure 10 and Figure S3 show that the droplets in Def-ARG are, on average, around 3-10 μm larger than in Mod-ARG, at least at the particular times we show.

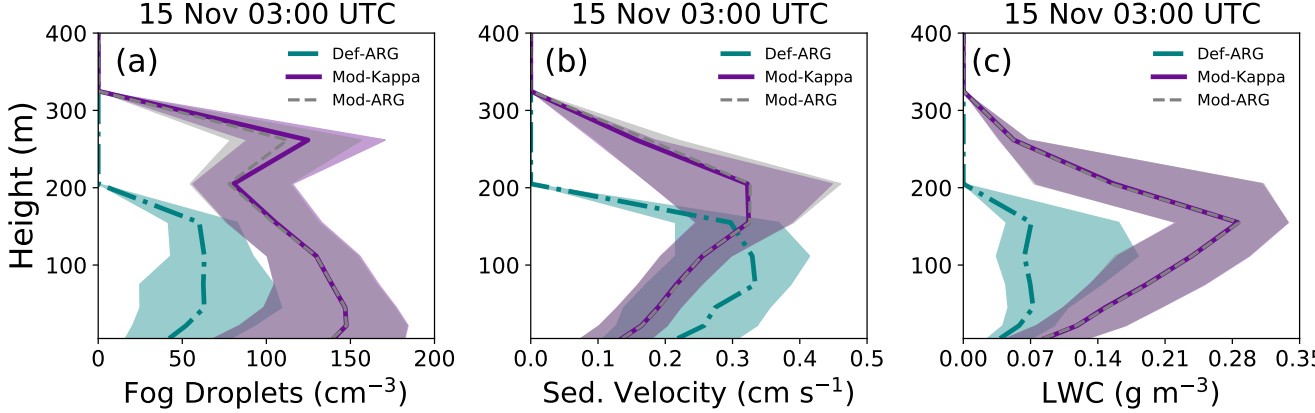

**Figure 14.** Vertical profiles of $N_d$ (a), droplet sedimentation velocity (b), and LWC (c) in fog for the 15th November ParisFog case (at 03:00 UTC) for different simulations.

In Figure 14, from the 500 m model, we show simulated vertical profiles of $N_d$, LWC and sedimentation velocity at 03:00 UTC for the 15th November fog case. The increase in LWC from Def-ARG to Mod-ARG in this case is fairly typical for this

increase in $N_d$; profiles from some other fog cases shown in supplementary Figures S13-S16 and are discussed in more detail in the companion paper. Where there is substantial fog, Def-ARG has higher sedimentation rates. However, other aerosol-cloud interactions mediated by radiation are also possible. For most fog cases, a higher LWC in Mod-ARG also leads to a higher fog top height, perhaps due to increased turbulence (also see Figure S16). A positive feedback mechanism can be proposed as follows: (a) reduced droplet sedimentation increases the liquid water path, enhancing cloud-top radiative cooling.

Radiative cooling may also be increased directly due to the smaller droplet effective radius, which leads to more efficient absorption of upwelling surface radiation that is subsequently re-emitted (Boutle et al., 2018). (b) Enhanced radiative cooling promotes further condensation of water vapor and intensifies turbulence (Boutle et al., 2018), resulting in a much optically thicker fog layer. (c) The increased turbulence then accelerates condensation through adiabatic cooling, raising the liquid water path and lifting the fog top, both of which contribute to even greater radiative cooling rates. While these processes are

very likely interrelated, this feedback mechanism remains speculative. More detailed models with finer grid resolution might more accurately represent possible counterbalancing effects, for example enhanced evaporation in the entrainment zone when sedimentation rates are reduced (Bretherton et al., 2007).

### 5.5 Evaluation of Fog Top Droplet Concentrations

In Figure 15, we compare simulated droplet concentrations at the top of the fog to retrievals from the Moderate Resolution

Imaging Spectroradiometer (MODIS) Terra satellite. The satellite retrievals have large uncertainties, but they do allow us



to evaluate (to some extent) the spatial variability of the droplet concentration across our model domain, which the SIRTA measurements do not.

We calculated $N_d$ using observations of cloud optical depth, cloud droplet effective radius, and cloud top temperature from MODIS, using the Collection 6 Level 2 dataset, and following equations $1 - 11$ described by Grosvenor et al. (2018). These
equations assume that the vertical profile of liquid water content is adiabatic, which is clearly erroneous in some of the fog cases (the vertical profiles of LWC and $N_d$ from a few ParisFog cases are shown in supplement Figures S11 and S13), but perhaps better than any other possible simple assumption. The fog top droplet concentrations can be calculated as:

$$N_d = \frac{\sqrt{5}}{2\pi k}\left(\frac{f_{ad}c_w\tau_c}{Q_{ext}\rho_w r_e^5}\right)^{\frac{1}{2}} \tag{4}$$

The constant $k$ is the ratio of the volume mean droplet radius to the effective radius, while $f_{ad}$ is the degree of adiabaticity.
We use $k = 0.8$ and $f_{ad} = 0.7$ following Grosvenor and Wood (2014). $c_w$ is the rate of increase in liquid water content with height for a moist adiabatic ascent. $\tau_c$ is the cloud optical thickness. $Q_{ext}$ is the extinction efficiency factor, set to 2 following Hulst and van de Hulst (1981). $\rho_w$ is the density of water.

We show the spatial variation of fog top droplet concentrations in the 500 m model from the MODIS satellite (subfigures b, e, h) and the MOD-Kappa simulation (subfigures c, f, i) for the ParisFog cases on November 15th 10:00 UTC, and on November
20th and 23rd at 11:00 UTC. We show MODIS data only for those locations where the cloud top height is below 1.5 km; higher cloud is the primary reason why subfigures b and e appear sparse. We also show the histogram of $N_d$ from the Def-ARG, and Mod-Kappa simulations and compare them with the MODIS-derived droplet distribution (subfigures a, b, c). We normalized the histograms to account for different numbers of foggy/cloudy gridboxes in different simulations. The selected days are the only days on which MODIS sees significant cloud with top height below 1500 m. Unfortunately, on November 23, our
evaluation at SIRTA showed that fog LWC is very poorly simulated by the model, so we do not expect good results. On the other two days, we have fewer valid retrievals, but still hundreds to thousands in total, enough to gain some insight into the $N_d$ variability.

Similar to the fog monitor measurements presented earlier, we find that the default simulation severely underestimates $N_d$. With updates to the ARG scheme (and hygroscopicity fixes), in simulation Mod-Kappa, the $N_d$ distributions are in much better
agreement with the observations (considering only 15th and 20th November). Both the mean and spatial variability appear to match reasonably well.

While the comparison of simulation and satellite data is valuable because it gives a sense of real observed spatial variability in $N_d$, the evaluation of simulated $N_d$ with MODIS-derived $N_d$ must be interpreted with care. Firstly, the satellite measurements have large uncertainties, estimated at 78% at the pixel level by Grosvenor et al. (2018) even for relatively adiabatic clouds.
Additionally, we cannot be sure our MODIS retrievals are of fog rather than cloud. The retrieved cloud top height used in both our 1.5 km threshold for 'likely fog' and in the $N_d$ retrieval itself is uncertain, and layers of cloud that is not fog may still have tops below 1.5 km. Indeed, we know that many of the fogs we study are obscured by higher cloud (although this is often simulated poorly by the UM).

**Figure 15.** Histogram of fog-top $N_d$ from Def-ARG and Mod-Kappa compared with $N_d$ observations from MODIS satellite (subfigures a, d, g). Spatial variation of $N_d$ from MODIS in the 500 m model domain is shown in subfigures b, e, h. The spatial variation of $N_d$ from the Mod-Kappa simulation in the 500 m model for different ParisFog cases is presented in subfigures c, f, i.





## 6  Discussions and Conclusions

The activation of fog droplets at the SIRTA observatory near Paris is studied with a weather and climate model. The aim is to examine how well aerosol and cloud microphysics designed for (relatively) computationally inexpensive weather and climate simulations can simulate droplet concentrations in fog. We simulated 11 fog events that were observed during the 2011 ParisFog field campaign, using a 500 m resolution setup. The aerosol microphysics scheme simulates reasonably realistic aerosol number concentrations and size distributions, and the nitrate aerosol scheme of Jones et al. (2021), tested here at kilometer-scale spatial

resolution for the first time, also seems to perform well. The key findings of our work are listed below:

■ **The model produces satisfactory simulations of fog cover, given that forecasting fog is difficult**

The model could produce some fog in our 500 m domain during all fog events observed at SIRTA, but the fog was not always co-located with the observations and there were some simulated fogs that did not correspond to observations.

■ **Droplet concentrations in the default setup are often biased low due to biases in parameterized aerosol activation**
**and probably poorly resolved turbulence**

Fog droplet concentrations were underestimated by our Def-ARG simulation, in which the model is run without modifications to the activation scheme. Sometimes, the underestimate is more than a factor of five, despite a slight underestimate of the aerosol number concentrations.

■ **Improving the ARG parameterization leads to a (usually) realistic simulation of droplet number concentrations**

We tested the ability of the ARG activation scheme to simulate aerosol activation scheme in fog-like conditions, and applied improvements to the parameterization we document separately. The updated activation parameterization is more physically realistic, droplet concentrations increase, and model performance improves. Similar improvements could be achieved by including a contribution to updraft speeds from sub-grid turbulence with a tuning factor. However, in our model, combining the update to the ARG parameterization with vertical velocities that match observations would almost
certainly lead to overestimates of $N_d$. Our modifications to hygroscopicity did not have much effect on $N_d$ in this model setup but will likely be useful elsewhere, for example in marine regions. The updates to hygroscopicity are included in Unified Model code releases from version 13.3 onward and we expect that they will be included in version 2 of the UK Earth System Model (UKESM2).

■ **Liquid water content can be simulated realistically some of the time and is sensitive to aerosol concentrations, but**
**large biases remain**

Fog LWC was generally realistic (within a factor of two of the observations during most of the fog events), although sometimes underestimated. Simulated surface liquid water content trends during the fogs only infrequently matched observations. The variation in LWC was similar to the variation in $N_d$ between our simulations. The LWC is sensitive to modifications to the activation parameterization that lead to substantial perturbations to the droplet concentration.



Simulating fog in a multipurpose model intended for operational weather forecasting and climate prediction has limitations compared to dedicated, detailed simulations designed to resolve large turbulent eddies (e.g. Mazoyer et al., 2017) or studies of fog with sectional cloud microphysics and prognostic supersaturation (e.g. Schwenkel and Maronga, 2019). In particular, we distinguish aerosols from fog droplets somewhat artificially and cannot simulate very large hydrated but unactivated aerosols with sizes well in excess of 1 μm. The loss rates of aerosol mass due to sedimentation may therefore be underestimated. Our grid spacing is too coarse to resolve all of the turbulence, and as we do not attempt to correct for this except in sensitivity studies, updraft speeds are likely underestimated. We are also unable to simulate bimodal droplet size distributions. However, the poor simulation of haze does not prevent relatively accurate simulations of fog droplet concentrations, with an activation parameterization that is tuned to a cloud parcel model, not to the observations with which we evaluate our UM simulations. In this paper, we assume (as in some other studies (e.g. Jia et al., 2019; Poku et al., 2019) but not all (e.g. Zhang et al., 2014; Mazoyer et al., 2017; Poku et al., 2021)) that radiative cooling does not activate droplets in radiation fogs, which is an obvious inadequacy we address in the companion paper.

These shortcomings, together with errors in the simulation of synoptic meteorology, may partly explain why our $N_d$ and LWC sometimes have large biases, sometimes we fail to simulate the occurrence of fog altogether, and sometimes we simulate fog where there is no fog in reality. Our case study from the ParisFog field campaign may be representative of moderately polluted urban fog but is unlikely to represent other types of fog, such as frontal fog or marine fog, where aerosol concentrations are likely lower and the surface fluxes driving fog formation are different. We also do not explicitly demonstrate in this paper that using prognostic aerosol number concentrations coupled to cloud microphysics actually improves simulated visibility for operational forecasting of fog, or that an aerosol climatology could not be used instead at lower computational cost, but we do present critical tests suggesting that these demonstrations may be possible in future. We refer the reader to Jayakumar et al. (2021) where a similar aerosol model without double-moment cloud microphysics was found to improve forecasts.

This paper shows that the microphysical properties of fog, such as droplet concentration and liquid water content, can be realistically simulated most of the time in a weather and climate model with prognostic aerosols running at 500 m grid resolution. In particular, we show that an activation parameterization designed to produce droplets in clouds via adiabatic cooling can yield realistic droplet concentrations in fog, and we isolate the sensitivities of the droplet concentration to key modeling choices and input parameters. Simulated fog droplet concentrations on the surface are especially sensitive to the activation parameterization. In our simulations at least, fog liquid water content is sensitive to the droplet number concentration, a result that provides some indirect support to other studies (e.g. Maalick et al., 2016) that suggest aerosols affect fog lifecycle. Our results lay the foundations for more detailed studies of the aerosol activation mechanism in fog in the companion paper. In the companion paper, we include a source of supersaturation from radiative cooling (together with a sink of supersaturation due to shortwave heating) in the same simulation case studies. We also investigate the performance of the model in simulating the vertical structure of fog using case studies from the UK LANFEX campaign in 2014 (Price et al., 2018). We examine how realistically the model simulates turbulence during LANFEX and calculate the relative importance of adiabatic and radiative cooling sources.





*Code and data availability.* All model and observation data used in this work is available at: https://doi.org/10.5281/zenodo.14004871 (Ghosh
et al., 2024a). The Terra/MODIS cloud L2 data sets were acquired from the Level-1 and Atmosphere Archive & Distribution System
(LAADS) Distributed Active Archive Center (DAAC), located in the Goddard Space Flight Center in Greenbelt, Maryland : https://ladsweb.
nascom.nasa.gov/ (Last access: Oct 23 2024). A copy of the MODIS satellite data used in this work is available in the same archive. A
copy of the SEVIRI satellite data used in this study is also available in the same archive. All atmospheric simulations used in this work
were performed using version 13.0 of the Met Office Unified Model (UM) starting from the GA7.1 configuration (Walters et al., 2019), and
also included version 7.0 of JULES.. The source code used in this study is free to use. However, software for this research is not publicly
available due to intellectual property copyright restrictions, but is available to signatories of the Met Office Software license. Full descriptions
of the software, including the specific configurations used in this study, can be found in the text of this article and in articles cited therein.
A number of research organizations and national meteorological services use the UM in collaboration with the Met Office to undertake
atmospheric process research, produce forecasts, develop the UM code, and build and evaluate Earth system models. To apply for a license
for the UM, go to https://www.metoffice.gov.uk/research/approach/modelling-systems/unified-model (last access: Oct 23 2024; Met Office,
2024), and for permission to use JULES, go to https://jules.jchmr.org (last access: Oct 23 2024). Rose and Cylc software were used to drive
the Unified Model. The simulations were run using Rose version 2019.01.3 and Cylc version 7.8.8 (and 8.3.3 for a couple of simulations),
which are publicly available at https://doi.org/10.5281/zenodo.3800775 (Shin et al., 2020), https://doi.org/10.5281/zenodo.4638360 (Oliver
et al., 2021), and https://doi.org/10.5281/zenodo.12801923 (Oliver et al., 2024) respectively. Both Rose and Cylc are available under v3 of
the GNU General Public License (GPL). The full list of simulation identifiers for the simulations in this paper is given below.

– Simulation Def-ARG: u-dh438

– Simulation Mod-ARG: u-dh848

– Simulation Mod-Kappa: u-dh847

– Simulation Def-ARG-Sigma: u-dk245

– Simulation Mod-ARG-Sigma: u-dk246

*Author contributions.* PG and HG formulated the idea of the paper with important contributions from all co-authors. PG and HG designed the
simulations and set up different model configurations with contributions from AJ. MM supplied the data from the ParisFog field campaign.
PG ran all the simulations. PG analyzed the simulation and observation data with contributions from NA and HG, and wrote the paper with
comments and suggestions from all co-authors.

*Competing interests.* The authors declare that they have no conflict of interest.

*Acknowledgements.* This research was supported by the U.S. Air Force Life Cycle Management Center (LCMC) collaboration with Oak
Ridge National Laboratory (ORNL). The computational resources on Air Force Weather HPC11 are provided by the Oak Ridge Leadership
Computing Facility (OLCF) Director's Discretion Project NWP501. The OLCF at Oak Ridge National Laboratory (ORNL) is supported by
the Office of Science of the U.S. Department of Energy under Contract No.DE-AC05-00OR22725. We thank the scientists responsible for





715   the ParisFog field campaign. Model simulations are material produced using Met Office software. This work used the Extreme Science and Engineering Discovery Environment (XSEDE), which is supported by the National Science Foundation Grant ACI-1548562. Specifically, it used the Bridges-2 system, which is supported by the NSF Award ACI-1928147, at the Pittsburgh Supercomputing Center (PSC). This work also used Bridges-2 at the PSC through allocation atm200005p from the Advanced Cyberinfrastructure Coordination Ecosystem: Services & Support (ACCESS) program, which is supported by National Science Foundation grants #2138259, #2138286, #2138307, #2137603, and

720   #2138296.



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
