# Peer review of "High sensitivity of simulated fog properties to parameterized aerosol activation in case studies from ParisFog"

_EGUsphere, 2024_

## Referee Comment (RC3)

The paper "High sensitivity of simulated fog properties to parametrized aerosol activation in case studies from Paris Fog" by Ghosh et al. addresses an important scientific question about "aerosol-cloud" interactions, which remains a major uncertainty in estimates of the indirect forcing exerted by aerosols. The paper is well written and the study objective and simulation design are clear and well suited to investigate the influence of aerosols in the case of fog.

More specifically, the authors investigate the ability of the UK Met Office Unified Model to simulate aerosols and fog properties during case studies from the ParisFog field campaign at the SIRTA site in November 2011. In particular, they explore the sensitivity of different fog properties to changes in simulated aerosol activation. This work is interesting, well presented, complete and uses several in-situ observations to evaluate the performance of different model configurations.

At this stage, I think that this study could be improved or completed in several aspects. The effects of certain biases on aerosol properties (number concentrations, chemical composition) or certain meteorological variables (as humidity) could be discussed more and are not detailed enough.

Major points :

First and for the Mod-Kappa test, a value of 0.1 is used for OC. Does it take into account SOA (which is not represented in the model) ? Does this value therefore represent a « low » value of Kappa that does not consider hydrophilic secondary organics aerosols ?

The time series of the number of foggy grid boxes (Figure 4) shows a large difference between the Def-ARG and Mod-Kappa simulations. Could this point be more detailed and discussed ? Is it possible to show the low cloud fraction for the two different simulations ?

With regard to the different evaluations discussed in sections 5.2 and 5.3, and even if the overall performance of the model is satisfactory, it would be interesting to discuss a little more the impact of certain biases on the simulated droplet concentration. For example, what is the possible effect of the (generally) negative bias on surface humidity for 15, 16, 17 or 24-26 November? What is the effect of the underestimation (sometimes with a significant bias, e.g. 21-22 and 23 November) of the aerosol concentration on the microphysical properties of clouds? For example, could the underestimation observed on 19, 21, 22 or 23 possibly explain part of the bias in simulated Nd (Figure 12e,f,h) ?

As mentioned previously and even if the hygrosocipity tests seem to be a second order effect, what is the possible effect of underestimating the contribution of SOA and sulphate hygroscopic aerosols (Figure 9 and Table 4), which could possibly affect the number of activated aerosols and fog droplets concentration ?

The comparisons of aerosol chemical composition are made with Crippa et al. (2013), but it could be interesting to use observed concentrations of inorganic aerosols at Sirta station to evaluate more chemical properties (if such in-situ observations exist) ?

Regarding the aerosol and droplet size distributions, it would be interesting to have a few more days to support the analysis shown in Figure 10 by indicating comparisons for days used to analyse the fog droplet concentration (Figure 12) or LWC (Figure 13). Here and as indicated in the text and Figure 10, the simulation of Nd by Def-ARG performs better than Mod-Kappa for the 15th day, whereas this is not the case for the 16th day. Is there any reason for this? it is not really discussed in the text. This could explain the overestimation of Nd by Mod-Kappa as shown in Figure 12a ?

Similarly, does the underestimation of the accumulation mode concentration on the 20th (Figure S3b) explain part of the negative bias in the simulated fog droplet concentration (Figure 12e) ? Could this point be discussed more in the context of the overall size distribution ?

In terms of droplet concentration and on the 24th (after 16:00) the Mod-Kappa simulation strongly overestimates the Nd observations while the aerosol concentration is very well reproduced by the model. Could this be due to some of the temperature biases indicated in Figure 6 or in the aerosol size distribution ?

In parallel with droplet concentration, it may be interesting to use the MODIS liquid cloud droplet effective radius product to constrain this microphysical property using the same method as proposed for Nd concentration.

---

## Author Comment (AC1)

**Response to Referee Comment 1 (Anonymous Referee #3)**

Accurately forecasting fog is currently a challenge for many operational numerical weather prediction models, with important implications in both weather and climate prediction. It has been shown in previous work (eg. Boutle et al 2018) that improper aerosol activation can lead to poor fog forecasts. This paper presents a detailed analysis of fog properties in model simulations as compared to observations for 11 fog events, and discusses the impact of aerosol activation as well as several modifications to the aerosol activation scheme. It is shown that the model, with 500m grid spacing, is capable of producing fog with reasonable droplet concentrations (Nd), and that Nd is sensitive to the aerosol activation parameterization. The authors include a discussion of the limitations of their study. Recommendation: Accept with minor revisions

We sincerely thank the reviewer for their helpful and encouraging comments. Below, we provide our detailed responses, with the reviewer's comments shown in Blue and our replies in Black.

During the process of addressing the reviewer's comments, we identified a bug in our code that affected the intended modifications in the Mod-Kappa simulation. After fixing this issue, we observed some quantitative differences in droplet concentrations between the Mod-ARG and Mod-Kappa simulations. However, the key take home message for the paper remains the same. We have updated the figures and text accordingly to reflect these corrections.

General Comments:

It may be helpful to the reader to include a table or graphic of the various schemes used in the various model components. I found the model description long and somewhat confusing, due to the 3 different model components, each using different parameterizations (just a suggestion).

In response to the comment, we have shortened the model description in the revised manuscript. We would like to clarify that our study includes both a global and a regional model setup, where the configurations are largely similar. However, to improve clarity, we have added a new table in the revised manuscript (Table 2) that outlines the important parameterizations used in the two setups.

Given that the Mod-ARG scheme significantly improves the results, how important are the multiple aerosol modes?

[Figure]

Figure R1: Droplet number concentrations (cm$^{-3}$) as a function of updraft speed for two different Aitken mode aerosol number concentrations: 0.05 cm$^{-3}$ and 50 cm$^{-3}$. Other conditions are same as Subfigure a,c of Figure 2 in the revised manuscript.

We agree it's important to consider whether the model is unnecessarily complex and expensive;

perhaps not all the modes are needed. The coarse mode concentrations, for example, are negligible in our simulations compared to the accumulation mode concentrations. In Figure 2 of the main text, we show that changing the coarse mode aerosol concentrations from $1 \, \text{cm}^{-3}$ to $0.1 \, \text{cm}^{-3}$ results in minimal changes in the activation fraction, so changing the coarse mode also does not affect the activation of the other modes. In Figure R1, we demonstrate how the total droplet concentrations changes when we decrease the Aitken mode aerosol concentrations from $50 \, \text{cm}^{-3}$ to $0.05 \, \text{cm}^{-3}$ using the updated ARG scheme (other conditions same as Subfigure a,c of Figure 2 in the revised manuscript). Similar to the coarse mode, we find minimal difference. Hence, in our simulations, if we did not have the Aitken or coarse mode aerosols, we would mostly likely get droplet concentrations similar to those shown in the manuscript.

In line 686 of the revised manuscript, we add the following:

"Further investigation could also clarify whether all five aerosol modes and full chemistry are needed to simulate fog droplet concentrations accurately,."

The role of sedimentation is only very briefly discussed, but could be important in removing droplets and aerosols. Was there any evaluation of the surface deposition due to sedimentation? Was there precipitation in any of the cases?

We agree with the reviewer that both gravitational sedimentation and interception by trees and buildings (fog deposition) are significant sinks for fog droplets. Gravitational sedimentation of droplets is incorporated into the model, and the relevant equations are described in the companion paper. We do not explicitly evaluate sedimentation rates; in principle, they could be obtained from the cloud radar (Delanoë et al., 2016) but to obtain them accurately from these observations would be a major undertaking. Moreover, the turbulent interception process is not included by default. In the companion paper, we address droplet interception using a proxy by increasing the sedimentation rate at the surface. We request the reviewer to kindly see Section 3.2 of the companion paper. To further clarify the treatment of sedimentation, in the revised manuscript, we have added the following statement in line 675:

"In our simulations, fog droplets can be lost to the surface via gravitational sedimentation; however, interception of droplets by trees and buildings near the surface is not included in the model by default. This process, which has been shown to play an important role in the fog lifecycle and vertical structure (Mazoyer et al., 2017), is addressed in the companion paper."

There was no precipitation during the 11 fog cases.

It is somewhat surprising to see a lack of sensitivity to aerosol hygroscopicity. Perhaps, as the authors suggest, that is due to the type of fog that is studied here.

While addressing the reviewer's comments, we discovered a bug in our code that affected the intended modifications in the Mod-Kappa simulation. After correcting this issue, we observed some quantitative differences in droplet concentrations between the Mod-ARG and Mod-Kappa simulations. However, these changes do not alter the main conclusions of our manuscripts. We have updated the figures and text accordingly, particularly in Section 4 of the revised manuscript, to reflect these corrections. The simulations presented in the companion paper also include this bug fix.

This study included a complex chemistry scheme, which is probably too expensive for typical weather and climate forecasting. Could the authors comment on the minimum requirements needed in NWP to produce reasonable fog simulations?

Indeed, the full chemistry scheme is computationally expensive for weather and climate forecasting

applications. However, as described in the manuscript, the inclusion of nitrate aerosols in our simulations necessitates the use of the full chemistry scheme, which would not be possible with a simplified approach.

In the UM, there is an option to use reduced chemistry, and now also a new option to use simplified aerosols where the composition is not tracked. Given that $N_d$ is relatively insensitive to aerosol composition, it should be feasible to simulate the fog lifecycle using these simplified schemes and/or aerosol climatologies. However, we have not tested this approach for the specific fog cases analyzed in this study. In line 686 of the revised manuscript, we added the following:

"Further investigation could also clarify whether all five aerosol modes and full chemistry are needed to simulate fog droplet concentrations accurately,"

The paper is long. I would encourage the authors to consider ways to shorten the text.

We thank the reviewer for their suggestion. Based on the reviewer's suggestion, we shortened the model description, and the introduction, removing a paragraph about saturation adjustment which we judged unnecessarily detailed. However, anonymous referee #2 (see Referee Comment 3) requested more explanation of the model biases. We added some text based on all the reviewers' suggestions, but ensured that we keep them short and concise.

Specific comments, typos, etc:

Line 51: thermodynamic is misspelled.

We have fixed the spelling in the revised manuscript.

Table 1: Welas 2020? Or WELAS-2000 as in lines 137 and 138?

We have changed 'welas' to 'WELAS'.

Line 166: The author should be van Weverberg, I believe.

We have fixed the citation in the revised manuscript.

Fig. 5 (and others): Please clarify if the model profile is a point value or an average over multiple points?

For Figure 5, we plot the T and RH profiles from the model at same location of the radiosondes. This is clarified in the revised manuscript. For plots shown later (with median and interquartile range), those are domain averages, as described in the manuscript.

Fig. 12 caption refers to S9, but it looks like that should be S10.

Yes. We have corrected the figure number (S11) in the revised manuscript.

Line 496: Are the 2 numbers referring to the two events on the 16th? Please clarify.

Yes, the numbers refer to two fog events on 16 November. We have clarified that in the revised manuscript.

Line 503: Subfigure is misspelled.

Spelling corrected.

Line 506: November is misspelled.

Spelling corrected.

Line 602: "subfigures (a,b,c)" should be subfigures (a,d,g).

Changed.

Fig. 15 (b, e, h): Retrievals is misspelled.

We have corrected the spelling.

In the supplemental material:

Fig. S2: Add to the caption the meaning of the shading (showing the observed foggy periods, presumably)

We modified the figure caption to include information about the shading during the observed foggy periods.

Figures S5 and S6 appear to be out of order.

The figures (S6 and S7 in the revised manuscript) are now in the correct order.

Fig. S8, line 2 of the caption: Omit either "boxes" or "gridboxes".

We have fixed the figure (S9 in the revised manuscript) caption.

Fig. S12: The legend in the figure doesn't match the experiment description in the caption.

We have fixed the figure (S13 in the revised manuscript) caption.

**Response to Referee Comment 2 (Anonymous Referee #1)**

Review of "High sensitivity of simulated fog properties to parameterized aerosol activation in case studies from ParisFog" by Ghosh et al.

This study presents simulations of multiple fog cases observed during the ParisFog field campaign. Updates to the aerosol activation scheme are shown to improve the simulation of droplet number concentration. These updates include more realistic hygroscopicities and updated parameters that control activation rate. These updated parameters are found to have a much larger impact than the hygroscopicities. The improved droplet concentration prediction does not necessarily improve the liquid water prediction. The improved Abdul-Razzak-Ghan scheme could potentially be used by others. The manuscript is long and could possibly be shortened. But otherwise, I don't have any major concerns with the study. It is a straightforward comparison of simulations with observations. The comparison is particularly comprehensive in this manuscript.

We sincerely thank the reviewer for their helpful and encouraging comments. Below, we provide our detailed responses, with the reviewer's comments shown in Blue and our replies in Black.

During the process of addressing the reviewer's comments, we identified a bug in our code that affected the intended modifications in the Mod-Kappa simulation. After fixing this issue, we observed some quantitative differences in droplet concentrations between the Mod-ARG and Mod-Kappa simulations. However, the key take home message for the paper remains the same. We have updated the figures and text accordingly to reflect these corrections.

It would be useful to mention at some point how droplet sedimentation to the surface is handled (if at all).

Kindly see our replies to referee comment #1 (anonymous referee 3). We agree that both gravitational sedimentation and interception by trees and buildings (fog deposition) are significant sinks for fog droplets. Gravitational sedimentation of droplets is incorporated into the model, and the relevant equations are described in the companion paper. We do not explicitly evaluate sedimentation rates; in principle, they could be obtained from the cloud radar (Delanoë et al., 2016) but to obtain them accurately from these observations would be a major undertaking. Moreover, the turbulent interception process is not included by default. In the companion paper, we address droplet interception using a proxy by increasing the sedimentation rate at the surface. We request the reviewer to kindly see Section 3.2 of the companion paper. To further clarify the treatment of sedimentation, in the revised manuscript, we have added the following statement in line 675:

"In our simulations, fog droplets can be lost to the surface via gravitational sedimentation; however, interception of droplets by trees and buildings near the surface is not included. This process, which has been shown to play an important role in the fog lifecycle and vertical structure (Mazoyer et al., 2017), is also addressed, albeit crudely, in the companion paper."

I don't think that section 5.5 adds much to the manuscript. Consider removing it.

We thank the reviewer for their feedback. We previously mentioned in the text (line 592 in the revised manuscript), "The satellite retrievals have large uncertainties, but they do allow us to evaluate (to some extent) the spatial variability of the droplet concentration across our model domain, which the SIRTA measurements do not." However, we failed to explain why this is important.

Throughout most of the paper, we validate the model only against ground-based observations. However, accurate representation of microphysical properties throughout the vertical structure of the fog is also important. To this end, we try to validate the $N_d$ at the fog-top by comparing the model results with MODIS observations. Although this comparison reveals spatial variability, the MODIS-derived $N_d$ are subject to large uncertainties (Grosvenor et al., 2018; Gryspeerdt et al., 2022). Comparison with surface observations suggests that the Def-ARG simulation almost always underestimates $N_d$; however, at the top of the fog it is not clear if Mod-ARG or Mod-Kappa is better than the default simulation or not. Contrasting model performance between surface and fog top further motivates us to better evaluate the vertical structure of fog against ground-based observations, an effort undertaken in the companion paper.

In Line 619 of the revised manuscript we add:

"We find that the spatial variability in $N_d$ is simulated realistically on both November 15 and 20. However, we also find that the spatial variability at cloud-top does not match the temporal variability at the surface (for example, the spatial standard deviation of cloud-top $N_d$ on 15 November from MODIS is 99 cm$^{-3}$ while the temporal standard deviation during the same fog event from the fog monitor is 38 cm$^{-3}$ at the surface). Therefore, the shaded ranges representing spatial variability we show on Figures 12 and 13 are only approximately comparable with the temporal variability in the observations. "

In line 630 of the revised manuscript, we add:

"However, the contrasting results near the surface and at the fog top, combined with uncertainties in satellite retrievals, further motivate the need to study the vertical structure of fog using ground-based observations—an effort addressed in the companion paper."

Figure 1 should have panel labels since they are referenced in the caption.

The panel labels have been added to the updated figure in the revised manuscript.

I believe that units should have spaces between components, e.g. on Line 265 ms$^{-1}$ technically reads as per millisecond, not meters per second. Use m s$^{-1}$ Same for gm$^{-3}$.

We have changed the units in the revised manuscript.

Figure 3 – presumably the temperatures are cloud top temperatures?

We agree, the temperatures are indeed fog top temperatures. We have updated the figure caption accordingly in the revised manuscript.

Line 671 – I wouldn't state that LWC can be realistically simulated "most of the time". Even previously in the conclusions the authors only state "some of the time".

We agree that while $N_d$ can be realistically simulated most of the time, LWC can be realistically simulated only some of the time. Hence, we replace the existing text with "generally be simulated with reasonable accuracy" on line 691 of the revised manuscript.

A few typos/extra words to note: Lines 51, 135 ("the point at which point" is awkward), 503, 662 (grammatical parallelism).

Fixed.

**Response to Referee Comment 3 (Anonymous Referee #2)**

The paper "High sensitivity of simulated fog properties to parametrized aerosol activation in case studies from Paris Fog" by Ghosh et al. addresses an important scientific question about "aerosol-cloud" interactions, which remains a major uncertainty in estimates of the indirect forcing exerted by aerosols. The paper is well written and the study objective and simulation design are clear and well suited to investigate the influence of aerosols in the case of fog. More specifically, the authors investigate the ability of the UK Met Office Unified Model to simulate aerosols and fog properties during case studies from the ParisFog field campaign at the SIRTA site in November 2011. In particular, they explore the sensitivity of different fog properties to changes in simulated aerosol activation. This work is interesting, well presented, complete and uses several in-situ observations to evaluate the performance of different model configurations. At this stage, I think that this study could be improved or completed in several aspects. The effects of certain biases on aerosol properties (number concentrations, chemical composition) or certain meteorological variables (as humidity) could be discussed more and are not detailed enough.

We sincerely thank the reviewer for their helpful and encouraging comments. Below, we provide our detailed responses, with the reviewer's comments shown in Blue and our replies in **Black**. Two other reviewers suggested shortening the text but this reviewer requested further clarification on several aspects. To address the feedback from all reviewers, we have provided additional clarifications in the responses below, while keeping the added text minimal in the revised manuscript.

During the process of addressing the reviewer's comments, we identified a bug in our code that affected the intended modifications in the Mod-Kappa simulation. After fixing this issue, we observed some quantitative differences in droplet concentrations between the Mod-ARG and Mod-Kappa simulations. However, the key take home message for the paper remains the same. We have

updated the figures and text accordingly to reflect these corrections.

First and for the Mod-Kappa test, a value of 0.1 is used for OC. Does it take into account SOA (which is not represented in the model)? Does this value therefore represent a "low" value of Kappa that does not consider hydrophilic secondary organics aerosols?

We thank the reviewer for raising this concern. Secondary Organic Aerosols (SOA) are indeed included in the model following Mann et al. (2010). The value of 0.1 is used following Schmale et al. (2018) and Fanourgakis et al. (2019). Although different $\kappa$ values can be assigned to hydrophobic and hydrophilic organics, because we do not track these separately in the model, we use a single value in this study. In the revised manuscript, we clarify this and in line 324, we add the following:

"we assign a $\kappa$ of 0.1 to OC (both primary and secondary organics)"

This $\kappa$ value of 0.1 is more representative of hydrophilic organics, whereas a value of 0.01 would be more appropriate for hydrophobic organics (see Table 1 caption in Fanourgakis et al. (2019)).

The time series of the number of foggy grid boxes (Figure 4) shows a large difference between the Def-ARG and Mod-Kappa simulations. Could this point be more detailed and discussed? Is it possible to show the low cloud fraction for the two different simulations?

We agree with the reviewer that the timeseries of foggy boxes shows a large difference between the Def-ARG and Mod-ARG simulations, particularly during the fog events on November 15 and 16. We tried to further investigate this. Ultimately, we think the most important mechanism by which fog droplet concentration affects simulated fog cover is sedimentation: when droplet concentrations are smaller, higher sedimentation velocities tend to result in the loss of fog water, reduced radiative effects, and eventually the clearing of the fog.

The mechanism is clearly more active on some days than others. We think the variability in the differences between simulations are mainly attributed to slightly higher inversion height in the model for some of these fog cases. For example, at 23:00 UTC, on November 14 the modeled inversion height is about 500 m while on November 19 it is about 350 m (see Figure 5a,f of the revised manuscript). As a result, for the 15 November early morning fog case, the feedback mechanisms that impact fog microphysics are more pronounced compared to the 20 November early morning fog case. In simulations Mod-ARG and Mod-Kappa, a higher $N_d$ combined with smaller droplet sizes results in a reduction in droplet sedimentation likely lowering the droplet sink to the surface. The increased LWC further amplifies radiative cooling and turbulence, promoting additional droplet formation, as detailed in the companion paper.

In Figure R2, we present the vertical profiles of droplet number concentration ($N_d$), droplet sedimentation velocity, and liquid water content (LWC) from fog events on 15, and 20 November. On November 15, the fog top height is about 300 m, but on November 20 it is about 150 m. Hence, on November 15, there are more possible feedbacks. Moreover, at the time we show here, the fog top height (and the number of foggy gridboxes) is similar across all simulations on 20 November, but on 15 November differences in fog top height are observed between simulations. In line 370 of the revised manuscript, we add the following:

"Differences in fog cover between Def-ARG and Mod-Kappa simulations arise via feedback mechanisms. For example, the higher modeled sedimentation velocity near the surface on 16 November in the Def-ARG simulation compared to the others leads to lower $N_d$ and liquid water content, contributing to the reduced number of foggy grid boxes compared to Mod-ARG (as discussed later

[Figure]

Figure R2: Vertical profiles of fog droplet number concentration ($N_d$), droplet sedimentation velocity and liquid water content (LWC) for three fog cases on 15, 20 November at 06:00 UTC from the 500 m model during the ParisFog campaign in 2011.

and shown in Figure 14 and supplementary Figure S14e). The differences are larger at the start of the simulation period, on 15 and 16 November, than towards the end. This may be due to the slightly higher altitude of the top of the inversion present during the first two fog events (next section) compared to most of the others. This higher inversion top could support enhanced feedback mechanisms (discussed later) that influence both $N_d$ and fog top height. Exceptionally, on 24 November, the inversion top is also elevated, but the inversion does not start at the surface, and the fog onset is simulated much too early."

In Figure R3 (copied as supplementary Figure S1), we show the timeseries of average cloud fraction (at the surface). from the 500 m resolution model for simulations Def-ARG, Mod-ARG and Mod-Kappa (only from the foggy gridboxes). In line 377 of the revised manuscript we add:

"In supplementary Figure S1 we show the timeseries of average cloud fraction (at the surface) from the 500 m resolution model for simulations Def-ARG, Mod-ARG and Mod-Kappa (only from the foggy gridboxes). The cloud fraction is consistently lower in Def-ARG compared to other simulations."

With regard to the different evaluations discussed in sections 5.2 and 5.3, and even if the overall performance of the model is satisfactory, it would be interesting to discuss a little more the impact of certain biases on the simulated droplet concentration. For example, what is the possible effect of the (generally) negative bias on surface humidity for 15, 16, 17 or 24-26 November? What is the effect of the underestimation (sometimes with a significant bias, e.g. 21-22 and 23 November) of the aerosol concentration on the microphysical properties of clouds? For example, could the underestimation observed on 19, 21, 22 or 23 possibly explain part of the bias in simulated Nd (Figure 12e,f,h)?

We thank the reviewer for raising this concern. In Figure 6, we show the mean temperature and RH of the domain and compare it with point observations at different locations. Such a comparison has

[Figure]

Figure R3: Timeseries of average cloud fraction (at the surface) from the 500 m resolution model for simulations Def-ARG, Mod-ARG and Mod-Kappa (only from the foggy gridboxes). Foggy periods in the observations are shown in shaded grey. Tick marks on the x-axis are at midnight UTC time.

representativeness uncertainties. Therefore, to address this, we have updated Figure 6 in the revised manuscript and included green shaded regions showing the variability within ±2 standard deviations of the mean temperature and relative humidity. It can be seen that the observed RH is within the variability across the model domain on most days, though biases on some days, for example, 25 November, are still large. Although RH biases do not seem well correlated to biases in fog onset or dissipation time, temperature biases can be seen to correspond, for example on 21 November, the simulated temperature is too low and a fog is predicted when no fog was present at SIRTA. Moreover, at higher altitudes, the biases in temperature and humidity can be much larger, leading to differences in cloud cover above the fog top (not shown). In line 411 of the revised manuscript, we add the following:

"While observed RH usually falls within the modeled variability, substantial biases remain on certain days (e.g., 25 November). However, these RH biases are comparable to other numerical weather prediction simulations of these cases (Menut et al., 2014) and do not align consistently with errors in the onset or dissipation timing of fog. In contrast, temperature biases appear to be more influential - for example, on 21 November, the model underestimates the surface temperature, resulting in a false fog prediction at SIRTA. Moreover, at higher altitudes, the biases in temperature and RH can be much larger, leading to differences in cloud cover above the fog top."

The biases in the simulated $N_d$ can certainly be affected by biases in aerosol concentrations. As noted by the reviewer, CCN-like aerosols are somewhat underestimated on 19, 21, 22 and 23 November. Lower aerosol concentrations can definitely lead to lower $N_d$. In line 487 of the revised manuscript, we add the following:

"Aerosol concentrations are also somewhat underestimated for fog cases between 19[th] and 23[rd] November, which may have contributed to the lower simulated values of $N_d$."

As mentioned previously and even if the hygroscocipity tests seem to be a second order effect, what is the possible effect of underestimating the contribution of SOA and sulphate hygroscopic aerosols (Figure 9 and Table 4), which could possibly affect the number of activated aerosols and fog droplets concentration?

We agree that the impact of changes in hygroscopicity is relatively minor compared to that of the

revised ARG scheme, even after the bug fix. However, we would like to clarify that Figure 9 and Table 4 (now Table 5) do not represent a direct comparison between the model outputs and observations at the same time as the simulated fog events. Rather, they provide an approximate assessment of the model performance using measurements from a different time period.

If the model underestimates the total mass of sulfate and organics, this would likely lead to lower droplet concentrations and smaller droplet sizes (Zhang et al., 2014). However, if the deficit in the sulfate mass is compensated by an increase in nitrate, the impact on the droplet properties may be minimal. Furthermore, given that aerosol number concentrations are in good agreement with observations, and our results show a relatively weak sensitivity of $N_d$ to variations in aerosol hygroscopicity, we believe that moderate changes in sulfate and organics mass are unlikely to significantly affect droplet concentrations. In line 441 of the revised manuscript we add:

"Since aerosol number concentrations align well with observations and $N_d$ is relatively insensitive to hygroscopicities (as demonstrated later), the relative underestimation of sulfate and organic aerosols is unlikely to significantly impact $N_d$. "

The comparisons of aerosol chemical composition are made with Crippa et al. (2013), but it could be interesting to use observed concentrations of inorganic aerosols at Sirta station to evaluate more chemical properties (if such in-situ observations exist)?

We thank the reviewer for their suggestion. However, unfortunately, as far as we know, there are no in-situ observations of concentrations of inorganic aerosols at SIRTA during the period of our study.

Regarding the aerosol and droplet size distributions, it would be interesting to have a few more days to support the analysis shown in Figure 10 by indicating comparisons for days used to analyse the fog droplet concentration (Figure 12) or LWC (Figure 13). Here and as indicated in the text and Figure 10, the simulation of Nd by Def-ARG performs better than Mod-Kappa for the 15th day, while this is not the case for the 16th day. Is there any reason for this? It is not really discussed in the text. This could explain the overestimation of Nd by Mod-Kappa as shown in Figure 12a?

We agree that showing aerosol and droplet size distributions for only a couple of days is insufficient to fully support the analysis. In Figure S3 of the submitted preprint we had shown size distributions during two fog events. Based on the reviewer's suggestion, we show two more fog events in Figure S4 of the revised manuscript. These supplementary results further support the discussion in the main manuscript, as mentioned in Line 447 of the revised manuscript: "Similar plots at the same time on 18, 20, 22, and 26 November are shown in Figure S4.", and in line 448 of the revised manuscript: "On the days shown in Figure 10 and Figure S4, the model overpredicts the aerosol concentrations in the Aitken insoluble mode.", and in Line 568 we add: "Figure 10 and Figure S4 show that the droplets in Def-ARG are, on average, around 3-10 µm larger than in Mod-ARG, at least at the particular times we show."

We agree that the contrasting model performance in the Def-ARG simulation between 15[th] and 16[th] November is indeed noteworthy. This difference can be primarily attributed to variations in the simulated meteorology. On 16 November, the top of the modeled inversion layer is higher than on 15 November (see Figure 5a,b of the revised manuscript), allowing these proposed feedback mechanisms to develop further, as shown in Figure 14 and Supplementary Figure S15. Furthermore, the sedimentation velocity in the Def-ARG simulation on 16 November is higher ($0.3\,\mathrm{cm\,s^{-1}}$) compared to 15 November ($0.1\,\mathrm{cm\,s^{-1}}$), resulting in a higher droplet sink. Consequently, we observe significantly lower $N_d$ and a relatively poor model performance in Def-ARG on 16 November, despite the fact that the modeled aerosol concentrations were largely similar on both days.

In line 581 of the revised manuscript, we add: "On 15 and 16 November, the fog coverage in Def-

ARG is much lower compared to other simulations, as discussed earlier. However, on 16 November, the inversion height is slightly higher than on 15 November, allowing these proposed feedback mechanisms to develop further. Furthermore, the droplet sedimentation velocity in the Def-ARG simulation on 16 November is higher (0.3 cm s$^{-1}$) compared to 15 November (0.1 cm s$^{-1}$), resulting in a larger sink of droplets. Hence, we find that on 16 November the difference in fog top height between simulations is much larger compared to 15 November, and the default simulation severely underestimates the droplets on the 16th November case."

Similarly, does the underestimation of the accumulation mode concentration on the 20th (Figure S3b) explain part of the negative bias in the simulated fog droplet concentration (Figure 12e)? Could this point be discussed more in the context of the overall size distribution?

As shown in Supplementary Figure S4d (and also in Figure 8) of the revised manuscript, aerosol concentrations are indeed underestimated on the 20th November. In the Def-ARG simulation, we observe a significant underestimation of $N_d$, which could be at least partially attributed to low aerosol concentrations in the accumulation mode, as the reviewer suggests. In all simulations, aerosol concentrations are generally comparable; however, the Mod-Kappa simulation shows an increase in $N_d$ and a smaller negative bias relative to Def-ARG. This highlights the sensitivity of droplet activation to the ARG scheme and the role of updated aerosol hygroscopicities. In addition to that, the modeled temperature near the surface near the surface and the inversion height have large biases (as already discussed in the main text), which could also impact the concentration of $N_d$.

In line 485 of the revised manuscript, we add the following:

"However, the severe underestimation on 20$^{th}$ November could be attributed to a negative bias in aerosol concentrations in the accumulation mode (by about a factor of 2, as shown in Figure S3d) and somewhat large temperature biases (2-3°C. Aerosol concentrations are also somewhat underestimated for fog cases between 19$^{th}$ and 23$^{rd}$ November, which may have contributed to the lower simulated values of $N_d$. "

In terms of droplet concentration and on the 24th (after 16:00) the Mod-Kappa simulation strongly overestimates the Nd observations while the aerosol concentration is very well reproduced by the model. Could this be due to some of the temperature biases indicated in Figure 6 or in the aerosol size distribution?

We thank the reviewer for raising this important point. In the revised manuscript, we show that on the 24th (subfigure j of Figure 12), the Mod-Kappa simulation (with the bug fixes applied) shows much better agreement with the observations compared to Def-ARG and Mod-ARG between 16:00 UTC and 18:00 UTC. $N_d$ in Def-ARG is zero, and Mod-ARG overestimates by a factor 2. However, after 18:00 UTC, dissipation of the fog is not very well simulated in simulations Mod-ARG and Mod-Kappa. We agree with the reviewer that the aerosols in the model are in excellent agreement with the observations on November 24, and the biases in $N_d$ could be due to biases in meteorology. In Figures 5 (k) and 6 (a) of the revised manuscript, we show that the temperature at the surface is about 2-3°C higher in the model compared to the observations. Such high-temperature biases are likely to inhibit fog formation; however, these biases do not change among simulations (see Figure 5 and R4). Additionally, the differences in accumulation mode aerosol size distribution is minor among different simulations.

[Figure]

Figure R4: Vertical profiles of temperature and relative humidity from 500 m grid resolution model of simulation Mod-Kappa. The results are compared with the data from radiosondes launched from Trappes around 23:00 UTC on 14[th] to 25[th] November.

[Figure]

Figure R5: Vertical profiles of fog droplet number concentration ($N_d$), droplet sedimentation velocity and liquid water content (LWC) at 19:00 UTC on November 24 from the 500 m model during the ParisFog campaign in 2011.

To further investigate this, Figure R5 shows the vertical profiles of $N_d$, LWC, and droplet sedimentation rate at 19:00 UTC on November 24. Near the surface, the LWC is similar across simulations at approximately $0.001 \, \mathrm{g \, m^{-3}}$. However, the sedimentation velocity differs, with Def-ARG exhibiting the highest value at $0.09 \, \mathrm{cm \, s^{-1}}$, followed by Mod-Kappa at $0.07 \, \mathrm{cm \, s^{-1}}$, and Mod-ARG at $0.06 \, \mathrm{cm \, s^{-1}}$. These variations in near-surface sedimentation velocity may have contributed to the differences in $N_d$ observed among the simulations at 19:00 UTC.

In Line 363 of the revised manuscript, we add the following: "For the fog cases on 24[th] November, the timing of dissipation is either too early or too late across the simulations. For the second fog case

on this day (Subfigure j), the Mod-Kappa simulation shows better agreement with observations up to 18:00 UTC, where Mod-ARG overestimates $N_d$ by a factor of 2. Differences in droplet sedimentation velocity (discussed later) may have contributed to the variations in $N_d$. "

In parallel with droplet concentration, it may be interesting to use the MODIS liquid cloud droplet effective radius product to constrain this microphysical property using the same method as proposed for Nd concentration.

We agree that it would indeed be valuable to compare the MODIS-derived droplet effective radius or liquid water path with the model. However, one of the reviewers recommended that this section be completely removed. To balance the suggestions of both reviewers, we have decided to retain this section in the manuscript, but not to expand it with additional analysis in this paper. Instead, we will explore these comparisons in detail as part of a dedicated future study.

**References**

Delanoë, J., Protat, A., Vinson, J.-P., Brett, W., Caudoux, C., Bertrand, F., du Chatelet, J. P., Hallali, R., Barthes, L., Haeffelin, M., and Dupont, J.-C. (2016). Basta: A 95-ghz fmcw doppler radar for cloud and fog studies. Journal of Atmospheric and Oceanic Technology, 33(5):1023 – 1038.

Fanourgakis, G. S., Kanakidou, M., Nenes, A., Bauer, S. E., Bergman, T., Carslaw, K. S., Grini, A., Hamilton, D. S., Johnson, J. S., Karydis, V. A., et al. (2019). Evaluation of global simulations of aerosol particle and cloud condensation nuclei number, with implications for cloud droplet formation. Atmospheric chemistry and physics, 19(13):8591–8617.

Grosvenor, D. P., Sourdeval, O., Zuidema, P., Ackerman, A., Alexandrov, M. D., Bennartz, R., Boers, R., Cairns, B., Chiu, J. C., Christensen, M., Deneke, H., Diamond, M., Feingold, G., Fridlind, A., Hünerbein, A., Knist, C., Kollias, P., Marshak, A., McCoy, D., Merk, D., Painemal, D., Rausch, J., Rosenfeld, D., Russchenberg, H., Seifert, P., Sinclair, K., Stier, P., van Diedenhoven, B., Wendisch, M., Werner, F., Wood, R., Zhang, Z., and Quaas, J. (2018). Remote sensing of droplet number concentration in warm clouds: A review of the current state of knowledge and perspectives. Reviews of Geophysics, 56(2):409–453.

Gryspeerdt, E., McCoy, D. T., Crosbie, E., Moore, R. H., Nott, G. J., Painemal, D., Small-Griswold, J., Sorooshian, A., and Ziemba, L. (2022). The impact of sampling strategy on the cloud droplet number concentration estimated from satellite data. Atmospheric Measurement Techniques, 15(12):3875–3892.

Mann, G. W., Carslaw, K. S., Spracklen, D. V., Ridley, D. A., Manktelow, P. T., Chipperfield, M. P., Pickering, S. J., and Johnson, C. E. (2010). Description and evaluation of glomap-mode: a modal global aerosol microphysics model for the ukca composition-climate model. Geoscientific Model Development, 3(2):519–551.

Mazoyer, M., Lac, C., Thouron, O., Bergot, T., Masson, V., and Musson-Genon, L. (2017). Large eddy simulation of radiation fog: impact of dynamics on the fog life cycle. Atmospheric Chemistry and Physics, 17(21):13017–13035.

Menut, L., Mailler, S., Dupont, J.-C., Laurant, O., Piriou, B., Haeffelin, P., Siour, M., Elias, T., Puygrenier, G., Colomb, A., Bresson, Y., Delbarre, O., and Augustin, P. (2014). Predictability of the meteorological conditions favourable to radiative fog formation during the 2011 parisfog campaign. Boundary-Layer Meteorology, 150(2):277–297.

Schmale, J., Henning, S., Decesari, S., Henzing, B., Keskinen, H., Sellegri, K., Ovadnevaite, J., Pöhlker, M. L., Brito, J., Bougiatioti, A., et al. (2018). Long-term cloud condensation nuclei number concentration, particle number size distribution and chemical composition measurements at regionally representative observatories. Atmospheric Chemistry and Physics, 18(4):2853–2881.

Zhang, X., Musson-Genon, L., Dupont, E., Milliez, M., and Carissimo, B. (2014). On the influence of a simple microphysics parametrization on radiation fog modelling: A case study during parisfog. Boundary-layer meteorology, 151:293–315.